# Bacterial recognition by PGRP-SA and downstream signalling by Toll/DIF sustain commensal gut bacteria in *Drosophila*

**Shivohum Bahuguna**[1¤a], **Magda Atilano**[1¤b], **Marcus Glittenberg**[1], **Dohun Lee**[1], **Srishti Arora**[1], **Lihui Wang**[1¤c], **Jun Zhou**[2], **Siamak Redhai**[2], **Michael Boutros**[2], **Petros Ligoxygakis**[1] *

1 Department of Biochemistry, University of Oxford, Oxford, United Kingdom, 2 German Cancer Research Centre (DKFZ), Division Signalling and Functional Genomics, BioQuant and Medical Faculty Mannheim, Heidelberg University, Heidelberg, Germany

¤a Current address: German Cancer Research Centre (DKFZ), Division Signalling and Functional Genomics, BioQuant and Medical Faculty Mannheim, Heidelberg University, Heidelberg, Germany
¤b Current address: Kennedy Institute of Rheumatology, University of Oxford, Headington, Oxford, United Kingdom
¤c Current address: Genetics, Evolution & Environment, Division of Biosciences, Faculty of Life Sciences University College London, London, United Kingdom
* petros.ligoxygakis@bioch.ox.ac.uk

**Data Availability Statement:** All relevant data are within the manuscript and its Supporting Information files.

## Abstract

The gut sets the immune and metabolic parameters for the survival of commensal bacteria. We report that in *Drosophila*, deficiency in bacterial recognition upstream of Toll/NF-κB signalling resulted in reduced density and diversity of gut bacteria. Translational regulation factor 4E-BP, a transcriptional target of Toll/NF-κB, mediated this host-bacteriome interaction. In healthy flies, Toll activated 4E-BP, which enabled fat catabolism, which resulted in sustaining of the bacteriome. The presence of gut bacteria kept Toll signalling activity thus ensuring the feedback loop of their own preservation. When Toll activity was absent, TOR-mediated suppression of 4E-BP made fat resources inaccessible and this correlated with loss of intestinal bacterial density. This could be overcome by genetic or pharmacological inhibition of TOR, which restored bacterial density. Our results give insights into how an animal integrates immune sensing and metabolism to maintain indigenous bacteria in a healthy gut.

## Author summary

Gut bacteria (collectively called the bacteriome) have beneficial effects on the physiology of animals but how they are retained by the host is an open question. Here we report that the immune system of the fly recognises these bacteria and activates a metabolic pathway leading to the regulated breakdown of lipids. The latter seems to be important for retention of intestinal bacteria because when lipids stores accumulate, the number of intestinal bacteria that can be cultivated out of the fly gut is significantly reduced. In fly mutants with a reduced immune recognition or response, the TOR pathway, a major pathway for

**Funding:** The work was supported by European Research Council Consolidator Grant 310912 to PL. The funders played no role in the study design, data collection and analysis, decision to publish, or preparation of the manuscript.

**Competing interests:** The authors have declared that no competing interests exist.

metabolism and growth, inhibits lipid breakdown and is responsible for increased fat accumulation in the gut. Blocking this pathway (pharmacologically or genetically) restores both lipid levels as well as the density of cultivable gut bacteria to normal levels. Our results show that this interplay between immunity and metabolism with the regulation of lipid catabolism at its centre is important for the retention of the intestinal bacteriome.

## Introduction

The animal gut accommodates a diverse array of bacteria, which assist in regulation of digestion, supply of nutrients and metabolites as well as in immune development [reviewed in 1]. To reap benefits from these microbes, the host provides a symbiotic environment for sustaining them in the gut [2]. The *Drosophila* gut and its bacteriome is used as a simpler model to study such host-microbe interactions [2]. Although much less diverse compared to humans, the fly bacteriome is equally dynamic and changes with age and environmental conditions connected to reinfections during fly culture [reviewed in 3, see 4–9].

The *Drosophila* intestinal epithelium is immunocompetent and upon enteric infection initiates innate immune responses via the NF-κB pathway IMD, mediating the production of antimicrobial peptides (AMPs) as well as the pathway centred on Dual Oxidase, the enzyme needed for the generation of Reactive Oxygen Species (ROS) [10–12]. However, it also preserves commensal bacteria, since transcription of AMPs is suppressed by Caudal [13] and Nubbin [14], while bacterial-derived uracil is important for distinguishing pathogens from commensals [15].

The evolutionary conserved Target of Rapamycin (TOR) pathway is a major pathway controlling cellular metabolism and growth [reviewed in 16]. TOR balances lipid and glucose anabolism and catabolism in the cell through the activity of the TORC1 protein complex [16]. TORC1 promotes protein synthesis primarily through phosphorylation of the eIF4E Binding Protein (4E-BP/Thor) and p70S6 Kinase 1 (S6K1) [16]. 4E-BP is a translational inhibitor, which binds and inhibits the activity of eIF4E an eukaryotic translation initiation factor responsible for the recruitment of 40s ribosomal subunit at the 5'-cap of mRNA. Phosphorylation of 4E-BP lowers its affinity towards eIF4E [17]. This frees eIF4E, enabling it to promote cap-dependent translation [17]. In the case of S6K1, activated S6K1 promotes protein synthesis by activating inducers of mRNA translation initiation whilst degrading inhibitors [17].

*Drosophila* TOR has been extensively studied for its role in growth and development, using fly mutants or by treating flies with rapamycin [18–21]. Rapamycin treatment in stress conditions, led to upregulation of 4E-BP activity resulting in an increase of whole-fly lipid reserves that could be used for the long-term survival of these stress conditions [22]. In contrast, 4E-BP mutants were unable to preserve lipid stores and had thus compromised survival following starvation or oxidative stress [22]. More broadly, the consensus is that in both *Drosophila* and mice, 4E-BP regulates fat levels in stress conditions like starvation and oxidative stress [23]. During larval stages and when in food with poor nutritional value, the presence of the commensal *Lactobacillus plantarum* is important to sustain development through TOR, which is in turn crucial for sustaining this mutualistic relationship [24]. The metabolic state of the gut is also influenced by dietary conditions and in its turn influences the bacteriome. Diet-dependent adaptations of the microbiota require NF-κB-dependent control of the translational regulator 4E-BP and this where TOR and NF-κB "meet" [25].

*Drosophila* has three NF-κB proteins namely, Relish, Dorsal and the Dorsal-related Immunity Factor (DIF) [26]. DIF is downstream of the Toll signalling pathway [reviewed in 27]. Toll

and Toll-like receptor (TLR) signalling is one of the most important evolutionary conserved mechanisms by which the innate immune system senses the invasion of pathogenic microorganisms. Unlike its mammalian counterparts however, *Drosophila* Toll is activated by an endogenous cytokine-like ligand, the Nerve Growth Factor homologue, Spz [28]. Spz is processed to its active form by the Spz-Processing Enzyme (SPE) [29]. Two serine protease cascades converge on SPE: one triggered by bacterial or fungal serine proteases through the host serine protease Persephone [30–33] and a second activated by host receptors that recognise bacterial or fungal cell wall [32,33]. Prominent among these host receptors is the Peptidoglycan Recognition Protein-SA or PGRP-SA [34]. PGRP-SA binds to peptidoglycan on the bacterial cell wall without structural preference but depending on accessibility [35] and generates the downstream signal.

When the recognition signal reaches the cell surface, it is communicated intracellularly via the Toll receptor and a membrane-bound receptor-adaptor complex including dMyd88, Tube (as an IRAK4 functional equivalent) and the Pelle kinase (as an IRAK1 functional homologue) [36]. Transduction of the signal culminates in the phosphorylation of the IκB homologue, Cactus probably by Pelle [37], leaving the NF-κB homologue DIF to move to the nucleus and regulate hundreds of target genes including antimicrobial peptides (AMPs) [27].

In this study, we present evidence that PGRP-SA is important for the preservation of commensal intestinal bacterial density. Our results reveal that larvae and adults that are deficient in PGRP-SA or DIF have a significantly reduced commensal gut bacterial density. Inhibition of the activity of TOR by Rapamycin or TOR RNAi in enterocytes, restores bacterial density (but not diversity) in PGRP-SA or DIF mutant guts. However, flies mutants for PGRP-SA and deficient for 4EBP in enterocytes were unable to restore bacterial density upon Rapamycin treatment or TOR RNAi, demonstrating the important role of 4EBP. PGRP-SA mutants had increased intestinal fat stores that were restored to normal levels through Rapamycin or TOR-RNAi treatment in enterocytes. This restoration failed in PGRP-SA;4EBP double mutants indicating that 4EBP was crucial in regulating fat stores in the gut. Fat catabolism was important for gut bacterial restoration as flies deficient for PGRP-SA and treated with rapamycin were unable to restore bacterial density if the triglyceride lipase Brummer was knocked down in enterocytes. This mechanism gives an insight into how the host integrates immunity and metabolism to maintain commensal bacteria at the intestinal epithelium.

## Results

### Loss of PGRP-SA and DIF changes cultivable bacterial density in the *Drosophila* gut

The bacterial recognition protein PGRP-SA has been studied extensively for its interaction with Gram-positive bacteria (e.g. *Staphylococcus aureus*, *Streptococcus faecalis*, *Bacillus thuringiensis*, *Bacillus subtilis* and *Micrococcus luteus*) upon infection, but its potential association with commensal bacteria has not been explored. Hence, we wanted to investigate whether loss of PGRP-SA had any effect on intestinal bacteria. To this end, we used flies carrying the *PGRP-SA^seml^* mutation [34]. We analysed the cultivable gut microbial load of female *yw seml* and *yw* flies at different ages (larval stage, as well as 5 and 30-day old adult flies) (Fig 1A). We observed a significant decrease in the cultivable gut microbial load (log$_{10}$ Colony Forming Units or CFUs) of 3rd instar *yw seml* mutant larvae as compared to their *yw* genetic background larvae. A similar decrease was also observed in the guts of 5-day old *yw seml* adults. In contrast, 30-day old *yw seml* flies did not show a significant difference in their cultivable bacterial load compared to the genetic background (Fig 1A).

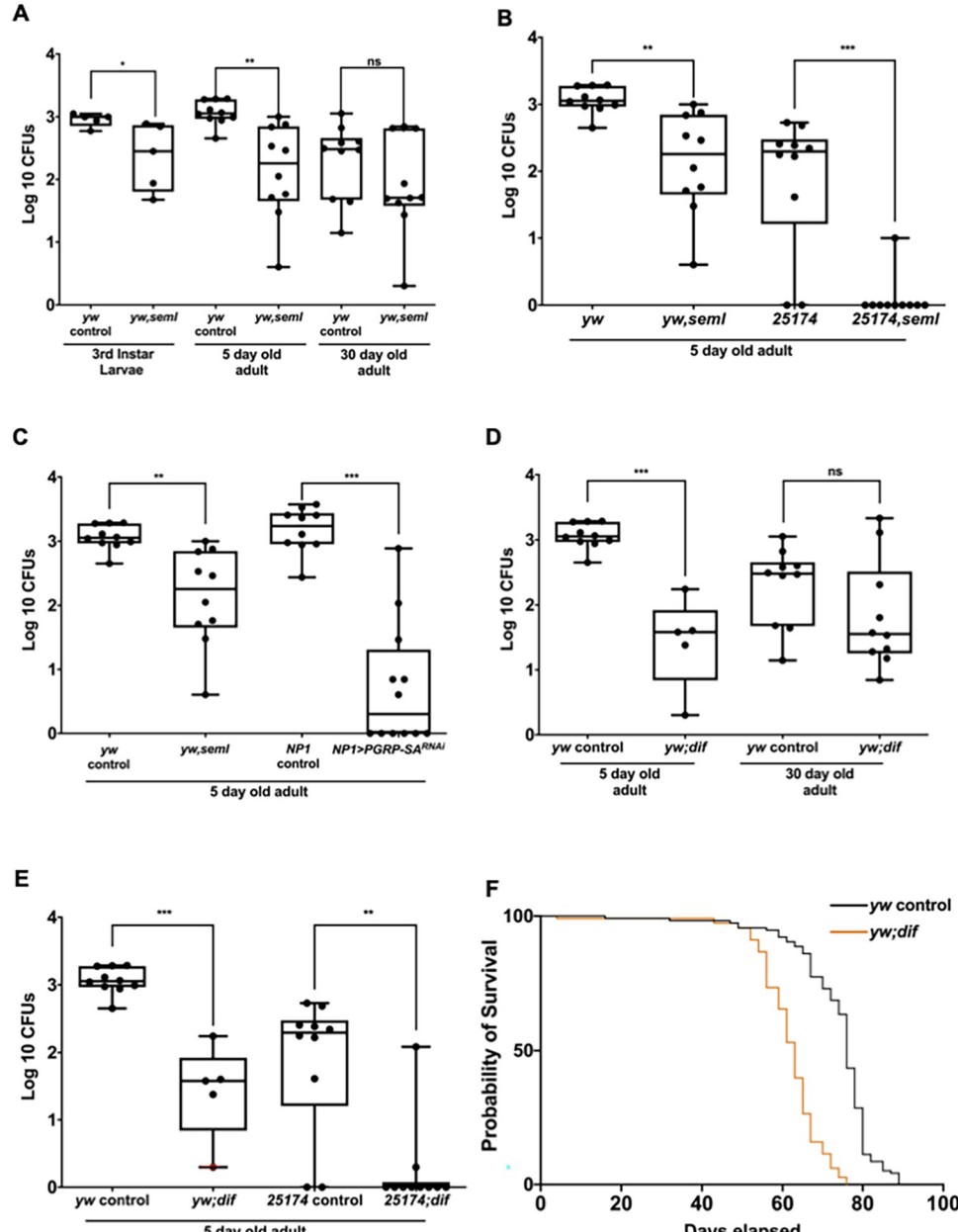

**Fig 1. Loss of the host receptor for bacteria PGRP-SA or the NF-κB homologue DIF reduces cultivable bacterial density in the gut.** **(A)** Larvae and 5-day old adults deficient for PGRP-SA have a significantly reduced density of cultivable bacteria while this is not the case for older *PGRP-SA^seml^* mutant flies (30-day old) flies. **(B)** This was not an effect of the genetic background as 5-day old *PGRP-SA^seml^* mutant flies in a DGRP line 25174 background also displayed reduced cultivable bacterial load. **(C)** The requirement for PGRP-SA was in enterocytes (ECs) as RNAi of PGRP-SA via the EC-specific NP1-GAL4 driver resulted in a significantly reduced cultivable bacteria load. **(D)** The *Drosophila* NF-κB ortholog DIF, downstream of PGRP-SA and the Toll receptor, was also important for the maintenance of bacterial density. **(E)** Enteric CFU reduction was not dependent on the genetic background since, as their age-matched *yw; dif* counterparts, the 25174; *dif* flies had also significantly reduced bacterial density. **(F)** The *yw; dif* flies had a reduced lifespan. Each dot represents one gut (n = 15 for both larvae and each adult category) in experiments done in three independent experiments (each experiment with n = 5). The log10 CFU values of mutants and controls were statistically compared using student's t-test (ns = not significant, *p < 0.05, **p<0.01, ***p<0.001, ****p<0.0001).

To exclude that load reduction of cultivable intestinal bacteria was due to the genetic background, we passed the *PGRP-SA^seml^* mutation through repeated backcrosses during 12 generations into the background of the 25174 strain. This is one of several fully sequenced inbred fly lines derived from a natural population established by the *Drosophila* Genetics Reference Panel (DGRP) [38]. Upon comparison, we found that 5-day old female DGRP *25174^seml^* flies also had a significantly reduced cultivable bacterial load as compared to their DGRP *25174* genetic background (Fig 1B). This reduction was in accordance with the observed differences in *yw PGRP-SA^seml^* mutant flies, confirming that the loss of cultivable bacteria was due to the absence of functional PGRP-SA and not due to the genetic background.

Commensal bacteria are compartmentalized in the mucosal layer of the gut, which consists of extracellular bacterial recognition proteins, AMPs and ROS secretions controlled by the gut epithelium [2]. Thus, we sought to assess if PGRP-SA secreted by enterocytes could influence the commensal bacterial load in the mucosal layer. We tested this by performing a cell-specific knockdown (RNAi) of PGRP-SA in enterocyte cells of adults using the GAL4/UAS system (in this case NP1-GAL4). Similar to the whole mutant *PGRP-SA^seml^* flies, enterocyte-specific *PGRP-SA^RNAi^* also resulted in a significant reduction in the load of cultivable gut bacteria (Fig 1C).

The presence of gut bacteria stimulates intestinal stem cell proliferation and division in *Drosophila* [39] and LGR5+ cells (which are progenitors of enterocytes) in mice [40]. Loss of bacterial load resulted in the reduction of Intestinal Stem Cells (ISCs) to Enteroblasts (EBs) in the gut of *PGRP-SA^seml^* flies (S1 Fig). ISC proliferation was stimulated significantly above wild type by a transgene expressing a constitutively active form of the Toll receptor (UAS-Toll10B) expressed in all progenitor cells and rescued ISC proliferation in *PGRP-SA^seml^* (S2 Fig). This showed that downstream signalling stimulated by Toll was able to rescue ISCs when PGRP-SA was non-functional.

To ascertain whether the transcription factor, which normally receives the PGRP-SA/Toll signal namely, the NF-κB homologue DIF was also involved in preserving the intestinal bacterial load of young flies we analysed guts from 5 and 30-day old *dif^1^* mutants. We observed that loss of DIF led to a significant reduction in the cultivable gut bacterial load of 5-day old female flies (Fig 1D). The same was observed when through 12 generations of backcrosses, the *dif^1^* mutation was incorporated into the DGRP *25174* genetic background (Fig 1E). Finally, a significant reduction in gut bacterial density was also the case when *dif* transcription was silenced through RNAi in enterocytes (S3A Fig). Of note, that these results contrast with what has been reported with loss of another *Drosophila* NF-κB protein namely, Relish where its loss did not influence intestinal bacterial CFUs in standard diets [25]. We have tested this by monitoring 16rRNA gene expression for two of the most represented genera in the *Drosophila* microbiome namely, *Acetobacter* and *Lactobacillus* in *Relish* mutant flies. We found that intestinal CFUs of these bacteria significantly *increased* in *rel^E20^* flies in comparison to heterozygous *rel^E20^*/+ or wild type controls (S3B Fig). Nevertheless, as in the case of *PGRP^seml^*, 30-day old *dif^1^* flies had a cultivable bacterial load in their gut comparable to their genetic background (Fig 1D).

Finally, longevity of *dif^1^* flies was significantly reduced compared to their genetic background (Fig 1F). A significant reduction in lifespan was also observed in *PGRP-SA^seml^* flies (S4A Fig). The loss of bacterial density in both *PGRP-SA^seml^* and *dif^1^* mutant flies suggested that members of the canonical Toll signalling pathway were involved in shaping the *Drosophila* gut microbiome in young flies. This was confirmed by looking at another member of the pathway, the Toll ligand Spz. Flies mutant for *spz* had significantly reduced intestinal CFUs (S4B Fig).

Given the effect on lifespan above, we were intrigued as to why reduction in the association of the host with cultivable commensal bacteria was only manifested in young flies. Was this an

effect that could be also found in older flies but was masked by the continued defaecation and re-infection taking place in the vials our flies were cultured in? To address this, we changed vials every 12 hours from day 1 of adulthood until day 30 with the view to reduce re-infection. The CFUs of *yw* and 25174 flies from day 1 to day 30 were reduced but stabilised indicating a stable state of association even at this rapid turnover of food vials (S5 Fig). In contrast, cultivable commensal bacteria in *PGRP-SA^seml* flies started from a lower effective bacterial population size compared to controls and showed a rapid reduction to significantly lower levels at day 30, indicating a bottleneck dependent on PGRP-SA in both the *yw* and 25174 genetic backgrounds (S5 Fig).

## Young PGRP-SA mutant flies lack *Lactobacillaceae*

The loss of cultivable bacteria prompted us to investigate the impact on the totality of intestinal bacteria including the non-cultivable component of the bacteriome. To this end, we measured by semi-quantitative PCR, the total amount of 16S bacterial rRNA gene in the gut of 5-day old flies and found a significant reduction in both *PGRP-SA^seml* and *dif1* compared to controls (S6 Fig). To test changes in diversity of the *PGRP-SA^seml* gut bacteriome we performed 16S rRNA gene high-throughput DNA sequencing. The relative abundance of bacterial families detected in the gut of female *yw* vs. *ywPGRP-SA^seml* flies (5 and 30 day old) are shown in Fig 2A. Guts of 5-day old female *yw* flies were dominated by *Lactobacillaceae* (48%) and *Acetobacteraceae* (23%) along with the presence of *Rhodobiaceae*, *Propionibacteriaceae* and other bacterial families (29%). Dominance of *Lactobacillaceae* in the gut of young flies correlated with previous

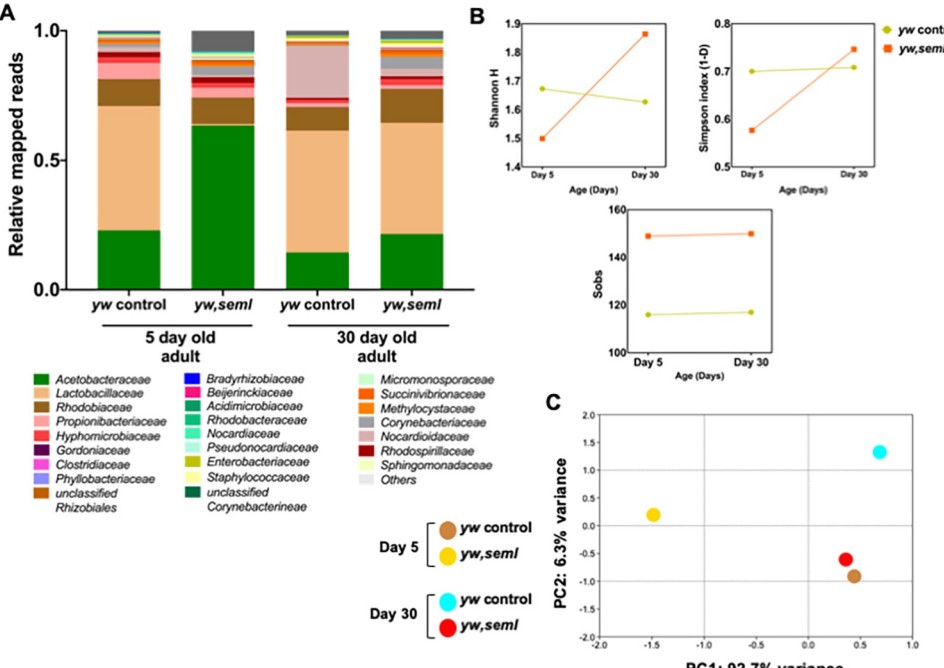

**Fig 2. Loss of PGRP-SA changes intestinal bacterial composition in young flies.** **(A)** The graph represents the relative abundance of bacterial families observed in the gut of 5-day and 30-day old female *yw*, *ywseml* flies revealed by 16S next-generation sequencing. The x axis represents y strains of different ages, and the y axis represents relative mapped reads. (n = 40 guts/strain). **(B)** The graphs represent alpha diversity indices of female yw and yw seml across two ages (5 and 30 days). (A) Simpson's (1-D) index, (B) Shannon H index an (C) Total number of bacterial families observed (Sobs). R represents biological repeat. (n = 40 guts/strain). **(C)** The graph represents PCA plot (Beta diversity) of female yw and yw seml across two ages (5 and 30 days). R represents biological repeat (n = 40 guts/strain).

studies [41]. In contrast to this, young 5-day old female *ywPGRP-SA^seml* mutant flies had a significantly reduced relative abundance of *Lactobacillaceae*, which accounted to only 0.6% of the total gut microbiome. This reduction in *Lactobacillaceae* was also observed in its biological repeat (S7 Fig). Furthermore, the relative abundance of *Acetobacteraceae* in *ywPGRP-SA^seml* mutant flies increased to 63% of the total gut microbiome and the remaining 36.4% encompassed other bacterial families (Figs 2A and S7).

Similar to their 5-day old bacterial composition, 30-day old female *yw* flies showed a dominance of *Lactobacillaceae* (47%) and *Acetobacteraceae* (15%). In contrast to their 5-day old siblings, 30-day old *ywPGRP-SA^seml* flies showed a substantial increase in *Lactobacillaceae* (43%), which turned out to be the most abundant family of the bacteriome followed by *Acetobacteraceae* (22%) (Fig 2A). The relative abundance of 30-day old *ywPGRP-SA^seml* flies was indistinguishable from 30-day old *yw flies*.

Statistical tests for Alpha and Beta diversity were performed to assess the diversity of the bacteriome within and between fly lines (Fig 2B). Approximately 116 bacterial families were recorded ($S_{obs}$) in young *yw* flies, which remained almost the same as the flies aged to 30 days ($\approx$ 117 families). Similarly, the number of bacterial families reported in young *ywPGRP-SA^seml* flies ($\approx$ 149 families) remained almost the same as *ywPGRP-SA^seml* flies aged to 30 days ($\approx$150 families) (Fig 2B). Alpha diversity measurements revealed that the 5-day old *ywPGRP-SA^seml* (Simpson 1-D = 0.5768, Shannon H = 1.499) were less diverse than *yw* flies (Simpson 1-D = 0.7004, Shannon H = 1.673). However, as the flies reached 30 days old, the bacteriome of *ywPGRP-SA^seml* flies (Simpson 1-D = 0.7465, Shannon H = 1.864) became more diverse than that of *yw* flies (Simpson 1-D = 0.7086, Shannon H = 1.627) (Fig 2B). Variation in the dominant *Acetobacteraceae* levels with age in *ywPGRP-SA^seml* mutant flies could be the reason for this diversity pattern. *Yw* flies showed no such change in the diversity pattern across these two ages as seen in Fig 2B. This could be due to a similar relative abundance pattern and almost same $S_{obs}$ between 5 and 30-day old flies.

Beta diversity analysis is performed to examine the differences in the bacterial composition between individual fly lines. Principal component analysis (PCA) is a tool to understand differences in multidimensional space and is commonly used to measure the beta diversity of microbial communities in *Drosophila* [7,42]. Beta diversity analysis (PCA plot) confirmed that *yw* and *ywPGRP-SA^seml* clustered far apart in both 5 and 30-day old flies, which indicated that they were dissimilar to each other. Interestingly, 30-day old *ywPGRP-SA^seml* flies appeared to be similar to their 5-day old genetic background (yw) as shown in the PCA plot (Fig 2C). This could be because both these flies share almost the same levels of the dominant families of *Acetobacteraceae* (*ywPGRP-SA^seml* 30-day = 22%, *yw* 5-day = 23%) and *Lactobacillaceae* (*ywPGRP-SA^seml* 30-day = 43%, *yw* 5-day = 48%).

## Genetic or pharmacological inhibition of TOR in *PGRP-SA^seml* flies restores density of cultivable gut bacteria

Intestinal bacterial density is dependent on host metabolism [43]. Gut bacteria thrive on the nutrients produced by digestion of the host's diet and intestinal secretions and are shaped by host-specific selective pressures such as the intestinal environment, food preference and eating habits [43]. We therefore tested the role of the major metabolic pathway TOR in regulating bacterial density in *ywPGRP-SA^seml* flies.

As illustrated in Fig 3A, silencing TOR via RNAi in enterocytes of 5-day old *ywPGRP-SA^seml* mutant flies resulted in a 10-fold increase in the cultivable microbial load as compared to untreated *ywPGRP-SA^seml*. To confirm, we conducted the symmetrical experiment where TOR was pharmacologically blocked with rapamycin. A similar result was observed, indicating

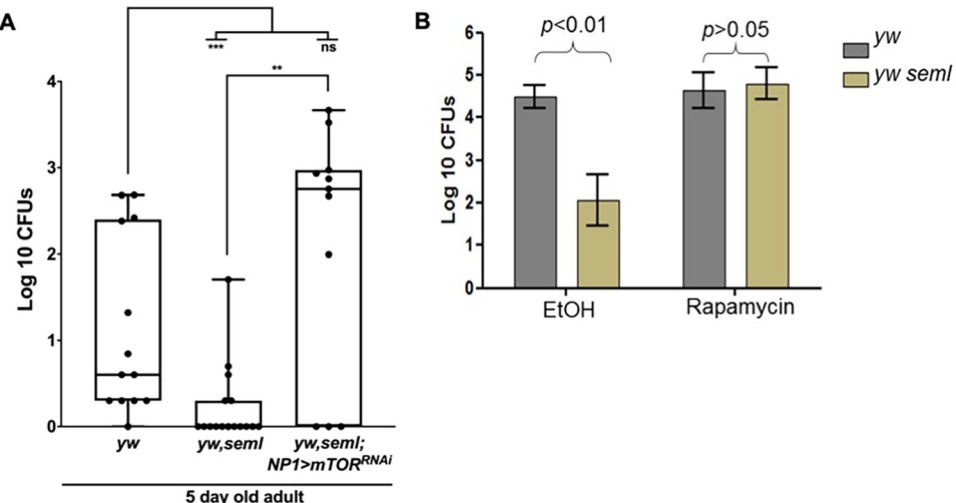

**Fig 3. In the absence of PGRP-SA, suppression of *Drosophila* TOR restores intestinal bacterial density. (A).** Silencing of TOR (via RNAi) in enterocytes of male *ywPGRP-SA^seml* flies or **(B)** pharmacological inhibition of TOR activity (via administration of rapamycin in the food) restored intestinal bacterial load. For panel A, each dot is an intestine (n = 15 for each category, total of three independent experiments) for panel B, each column is the median of three independent experiments (n = 20 for each experiment). Statistical comparisons were made using student's t-test (ns = not significant, **p<0.01, ***p<0.001).

restoration of the cultivable intestinal bacteria (Fig 3B). Furthermore, treatment with rapamycin restored the ability of the intestinal epithelium to renew with ISCs able to divide again (S8 Fig).

Nevertheless, restoration of bacterial density in young *ywPGRP-SA^seml* flies did not restore diversity (Fig 4A). Sequencing of 16S gut bacterial libraries of rapamycin-treated or dTOR^R-NAi^-treated *ywPGRP-SA^seml* flies, indicated that the absence of the *Lactobacillaceae* family was not restored. As shown in Fig 4A, rapamycin(+Rap) treatment of young (5-day old) female *yw* and *ywPGRP-SA^seml* flies resulted in a shift of their relative abundance. In *yw* flies, rapamycin treatment led to a 3-fold reduction of *Lactobacillaceae* (48% in *yw* vs. 17% in *yw* + Rap), 3-fold increase of *Acetobacteraceae* (23% in *yw* vs. 70% in *yw* + Rap) and a 10-fold decrease of *Propionibacteriaceae* (6% in *yw* vs. 0.6% in *yw* + Rap). Approximately 118 and 116 bacterial families (S$_{obs}$) were reported in rapamycin-treated and untreated *yw* flies respectively (Fig 4B, 5 days). Alpha diversity analysis revealed that treatment of *yw* flies with rapamycin reduced the diversity of the gut bacteria (*yw*: Simpson 1-D = 0.7004, Shannon H = 1.673; *yw* + *Rap*: Simpson 1-D = 0.4785, Shannon H = 1.107) (Fig 4C, 5 days). Dissimilarity of the bacterial composition between *yw* and *yw* +Rap was confirmed by analysing the PCA plot (Fig 4C). These results suggested that rapamycin reduced the diversity of bacterial composition in young *yw* flies.

As shown in Fig 4A, rapamycin treatment did not have a major effect on the relative abundance of microbial families in *ywPGRP-SA^seml* mutant flies. Both treated and untreated female *ywPGRP-SA^seml* flies lacked *Lactobacillaceae* (0.5% and 0.6% respectively) as compared to their genetic background (17% and 48% respectively). *Acetobacteraceae* largely dominated the gut bacterial population of both treated and untreated *ywPGRP-SA^seml* flies (59% and 63% respectively). Overall, 141 and 149 bacterial families (S$_{obs}$) were reported in rapamycin-treated vs. untreated *ywPGRP-SA^seml* flies respectively (Fig 4B). This showed that although inhibiting dTOR by Rapamycin restored bacteriome density in *ywPGRP-SA^seml* flies it did not influence bacteriome diversity.

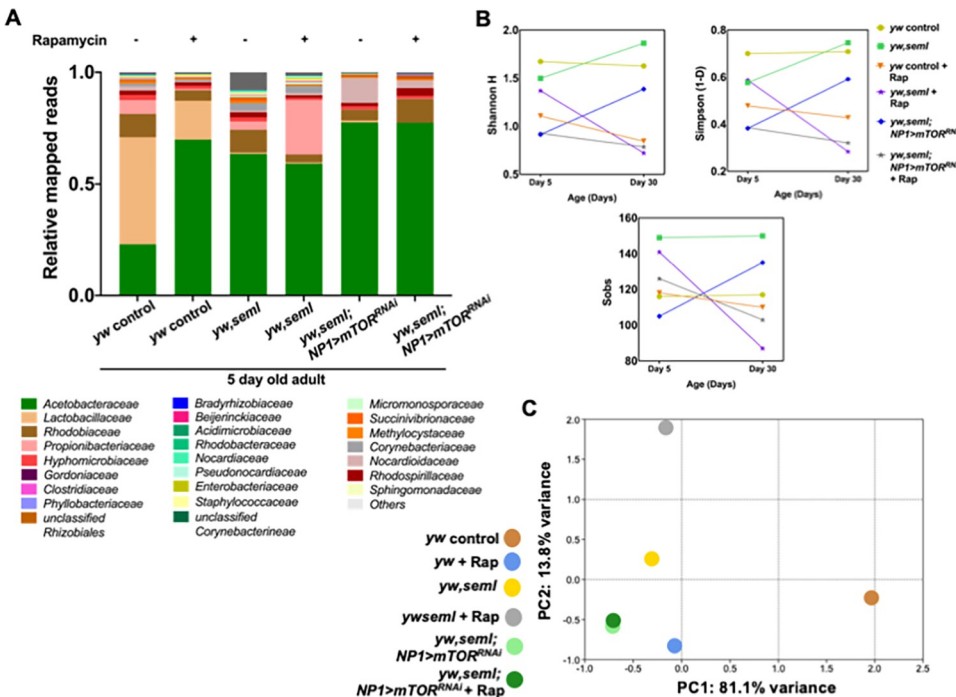

**Fig 4. Inhibition of *Drosophila* TOR does not restore bacterial diversity. (A)** The graph represents the relative abundance of bacterial families observed in the guts of 5-day old female *yw*, *ywseml* (females) and *ywseml; NP1GAL4>UAS-TOR^RNAi* flies (males) with (+) or without (-) rapamycin treatment revealed by 16S next-generation sequencing. The x axis represents y strains of different genotypes and treatments, and the y axis represents relative mapped reads. (n = 40 guts/strain/treatment). **(B)** The graphs represent alpha diversity indices of female *yw*, *ywseml* and *ywseml; NP1GAL4>UAS-TOR^RNAi* across two treatments (with or without rapamycin). (A) Simpson's (1-D) index, (B) Shannon H index an (C) Total number of bacterial families observed (Sobs). (n = 40 guts/strain/treatment). **(C)** The graph represents PCA plot (Beta diversity) of female *yw*, *ywseml* and *ywseml; NP1GAL4>UAS-TOR^RNAi* across two treatments (with or without rapamycin; n = 40 guts/strain/treatment).

Rapamycin treatment of 30-day old female *ywPGRP-SA^seml* flies resulted in a change in the gut microbial composition, which was mainly marked by a 4-fold increase of *Acetobacteraceae* (21% in *ywPGRP-SA^seml* to 85% in *ywPGRP-SA^seml* +Rap) and a 15-fold decrease of *Lactobacillaceae* (43% in *ywPGRP-SA^seml* to 3% in *ywPGRP-SA^seml* + Rap) (S9A Fig). Approximately 87 and 150 bacterial families (S_obs) were reported in rapamycin treated and untreated 30-day old *ywPGRP-SA^seml* flies respectively, suggesting a loss of diversity following rapamycin treatment. This was corroborated by alpha diversity analysis, which revealed that rapamycin treatment of 30-day old *ywPGRP-SA^seml* flies resulted in a decrease in diversity as compared to untreated controls. PCA plots further confirmed this result by placing rapamycin-treated and untreated *ywPGRP-SA^seml* flies far apart from each other, which suggested again that rapamycin influenced the microbial composition and diversity of 30-day old *ywPGRP-SA^seml* flies (S9B Fig). This result was contrary to 5-day old flies in which rapamycin treatment did not have a significant effect on either the composition or on the diversity of intestinal bacteria (S9C Fig).

When we compared the relative abundance of 5-day old *ywPGRP-SA^seml* female flies treated with rapamycin vs. *ywPGRP-SA^seml* with enterocyte-specific knockdown of dTOR (*ywPGRP-SA^seml*; *NP1>mTOR^RNAi*), we observed significant differences (Fig 4A). A dominance of *Nocardioidaceae* family (60%) was observed in *ywPGRP-SA^seml*; *NP1>mTOR^RNAi* flies, while dominance of *Acetobacteraceae* (84%) was seen in rapamycin-treated *ywPGRP-SA^seml* flies. Both fly lines lacked *Lactobacillaceae* in their gut (3% in *ywPGRP-SA^seml* + Rap versus 0.4% in

*ywPGRP-SA^{seml}; NP1>mTOR^{RNAi}*) (Fig 4B). The $S_{obs}$ for 30-day old rapamycin treated *ywPGRP-SA^{seml}* flies was ≈ 87 and the $S_{obs}$ for 30-day old *ywPGRP-SA^{seml}; NP1>mTOR^{RNAi}* was ≈ 135 families (Fig 4B). Alpha diversity measurements revealed that the gut bacteria of 30-day old *ywPGRP-SA^{seml}; NP1>mTOR^{RNAi}* flies (Simpson 1-D = 0.5917, Shannon H = 1.386) was more diverse than *ywPGRP-SA^{seml}* mutants fed with rapamycin (Simpson 1-D = 0.2836, Shannon H = 0.7219) (Fig 4B). Beta diversity analysis (PCA plot) showed that these two strains clustered far apart from each other, suggesting that they had a dissimilar composition based on the relative abundance of their bacterial families (Fig 4C). Major differences in the relative abundance and diversity in 30-day old *ywPGRP-SA^{seml}; NP1>mTOR^{RNAi}* as compared to 30-day old *ywPGRP-SA^{seml}*+ Rap could be due to the reduced levels of mTOR throughout most of the lifespan of the fly in the former vs. transient inhibition of the TORC1 complex by rapamycin in the latter.

## Restoration of bacterial density in *ywPGRP-SA^{seml}* requires the translation regulator 4E-BP

The fact that TOR inhibition restored bacterial density in *ywPGRP-SA^{seml}* mutants, pointed towards a potential metabolic shift in the gut under PGRP-SA/Toll-deficient conditions. Abrogating TOR activity may have reversed this or may have generated an alternative metabolic state that enabled bacterial growth again. Whatever the case, we reasoned that such a metabolic environment, which permitted normal gut bacterial density when both Toll and TOR signals were suppressed, required a regulator at the nexus of these two pathways. It has been recently shown that the *Drosophila* homologue of the mammalian 4E-BP, is regulating gut bacteria [25]. 4E-BP, an important downstream component of TOR signalling, includes several NF-κB binding sites [44].

These NF-κB sites respond to infections regulated by the Toll pathway [45]. Corroborating this point, we found that systemic *Candida albicans* infection, which activates the Toll response, also activated a 4EBP transcriptional reporter in enterocytes (S10 Fig). Quantitative real time PCR analysis showed a 4-fold decrease of 4E-BP in female *ywPGRP-SA^{seml}* flies and a 2-fold decrease in female *yw; dif^1* flies as compared to their genetic background (Fig 5A). Moreover, there was a significant increase in the phosphorylated form of 4E-BP when triggering the constitutively active Toll10B receptor in enterocytes (S11 Fig). These results indicated that the canonical Toll signalling pathway exerted control on 4E-BP not only after infection but also during homeostatic gut function.

Apart from being controlled by the host innate immunity, 4E-BP is also known to be regulated by mTORC1 [reviewed in 45]. Briefly, 4E-BP binds to translation initiation factor elF4E and prevents 5'cap-dependent mRNA translation. Activated mTORC1 phosphorylates 4E-BP and promotes translational protein synthesis. To test the effect of genetic as well as pharmacological inhibition of *Drosophila* TOR activity on 4E-BP levels in *ywPGRP-SA^{seml}* flies, we performed quantitative PCR. Results indicated a 3-fold increase in 4E-BP transcript levels upon rapamycin treatment and a 5-fold increase upon silencing TOR in the enterocytes of *ywPGRP-SA^{seml}* mutants as compared to 4E-BP levels in the gut of untreated *ywPGRP-SA^{seml}* (Fig 5B). Although a statistically significant increase in 4E-BP was observed in both cases, silencing of TOR through RNAi had more impact on the increase of 4E-BP transcript levels as compared to inhibition of TORC1 activity by rapamycin (Fig 5B).

The fact that restoring 4E-BP transcript levels by suppressing TOR correlated with the restoration of intestinal CFUs (Fig 3A and 3B), gave us reason to investigate whether 4E-BP was essential for mediating the restoration of cultivable bacterial load we had previously seen in young *ywPGRP-SA^{seml}* flies after rapamycin treatment. To do so, we compared

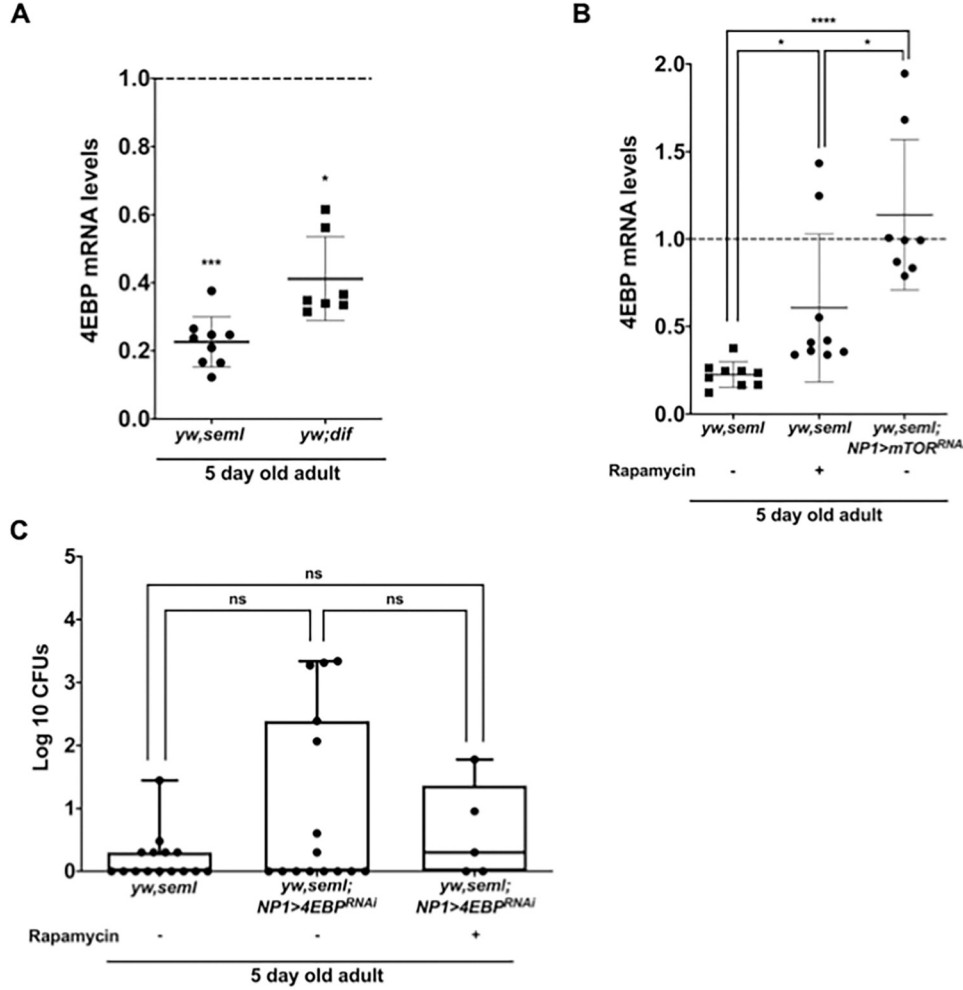

**Fig 5. 4EBP is involved in regulating gut bacterial density as the nexus of Toll/NF-κB and mTOR signalling. (A).** PGRP-SA and DIF influence *4EBP* mRNA levels in the *Drosophila* gut. RT-qPCR analysis of *4E-BP* gene expression in the guts of 5-day old female *yw seml and yw dif¹* flies relative to their genetic background (*yw*), which was set to 1 (dotted line; n = 10 guts/strain). **(B).** Inhibition of mTOR activity restored 4EBP levels in *seml* mutant guts. RT-qPCR analysis of 4E-BP expression in the guts of 5-day old female *yw seml*, female *yw seml* treated with rapamycin and female *yw seml; NP1>mTOR^RNAi* flies relative to *yw* female control, which was set to 1 (dotted line; n = 10 guts/strain). **(C).** 4E-BP mediates rapamycin-induced restoration of cultivable bacterial load in *yw seml* flies. The graph represents the cultivable gut bacterial load of 5-day old female *yw seml* (n = 10) and *ywseml/Y; NP1>4E-BP^RNAi* (n = 10) and *ywseml/Y; NP1>4E-BPRNAi* + rapamycin (n = 8) flies. In each case (A-C) the x axis represents different fly strains, and the y axis represents relative fold change calculated by ΔΔCT method. n = 3 biological repeats, (ns = not significant, *p<0.05, **p<0.01, ***p <0.001, ****p<0.0001).

*ywPGRP-SA^seml*; *NP1>4E-BP^RNAi* treated and untreated with Rapamycin, as well as untreated *ywPGRP-SA^seml* (Fig 5C). The bacterial load of untreated *ywPGRP-SA^seml* flies was statistically indistinguishable from untreated *ywPGRP-SA^seml*; *NP1>4E-BP^RNAi* (5C). Treating the latter with rapamycin did not restore bacterial density (Fig 5C). This was in contrast to what was observed previously for *ywPGRP-SA^seml* (Fig 3B). In turn, this suggested that rapamycin restored bacterial density in *ywPGRP-SA^seml* flies through 4E-BP and that 4E-BP was required in enterocytes (Fig 5C). Of note that just loss of *4E-BP* significantly decreases normal CFUs in the gut (S12 Fig). These results indicated that 4E-BP was pivotal for generating the metabolic environment that ensured normal bacterial growth downstream of PGRP-SA/Toll/NF-κB.

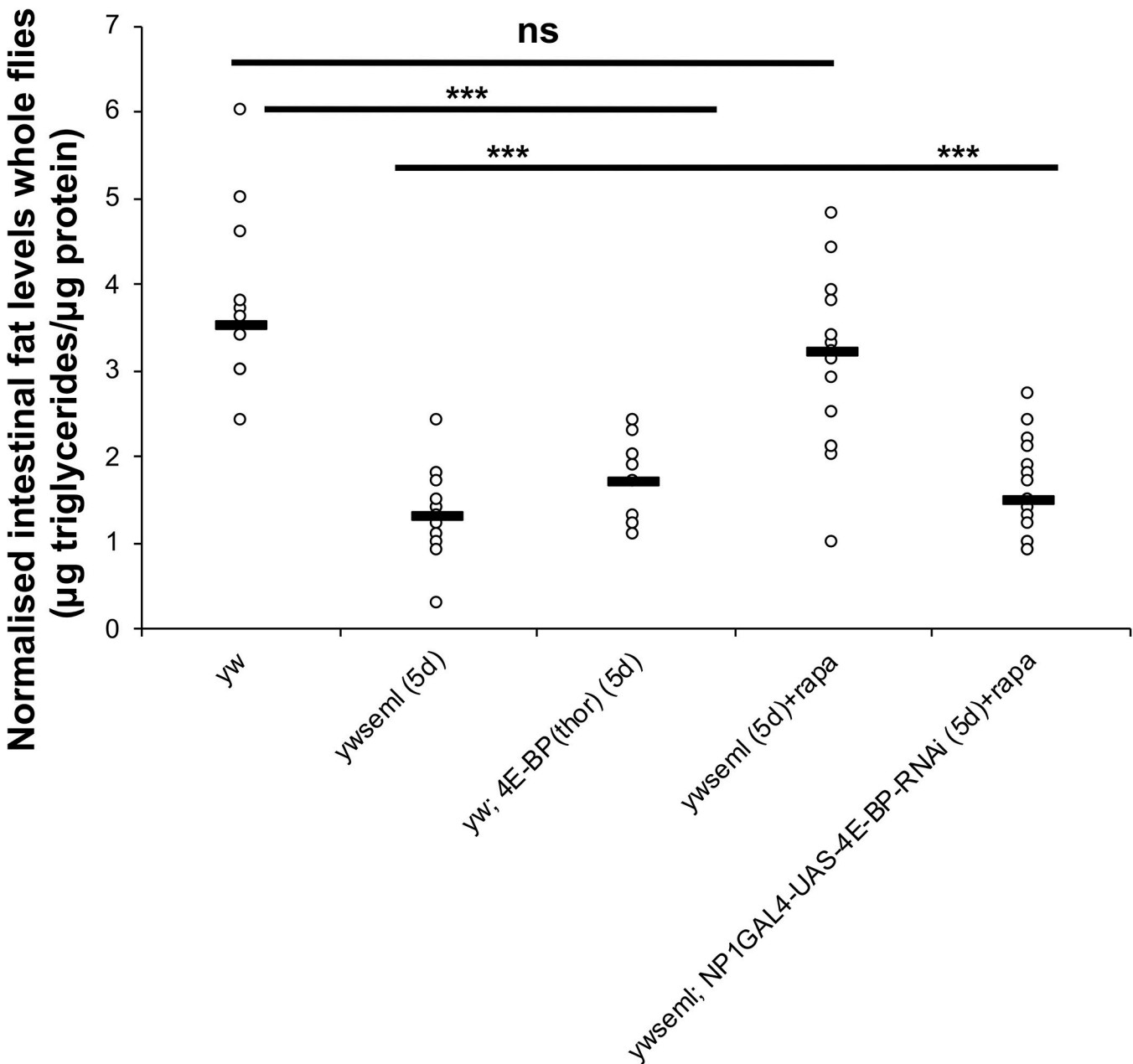

**Fig 6. Loss of PGRP-SA increases intestinal fat levels.** Loss of PGRP-SA increased intestinal triglyceride levels in 5-day old flies. This phenomenon was suppressed with pharmacological inhibition (rapamycin) or RNAi against TOR in ECs. This was dependent on 4EBP as *yw seml; NP1>4E-BP^RNAi* treated with rapamycin had fat levels statistically indistinguishable from *yw seml*. N = 15/genotype/treatment a total of three independent experiments (each with n = 5/genotype/treatment). Values of mutants and controls were statistically compared using student's t-test (***p<0.001, all other comparisons non-significant except *yw seml; NP1>4E-BP^RNAi* treated with rapamycin compared to *yw*, which has a p value<0.001-comparison not shown in the graph).

## The PGRP-SA/Toll/4E-BP axis regulates intestinal triglyceride stores

Reduction of bacterial density in 5-day old *PGRP-SA^seml* mutants was accompanied by an increase in intestinal triglyceride stores (Fig 6). In comparison, 5-day old *PGRP-SA^seml* mutant flies treated with the TOR inhibitor rapamycin showed a reduction of intestinal triglyceride levels (Fig 6). These were statistically indistinguishable from the levels of their *yw* genetic

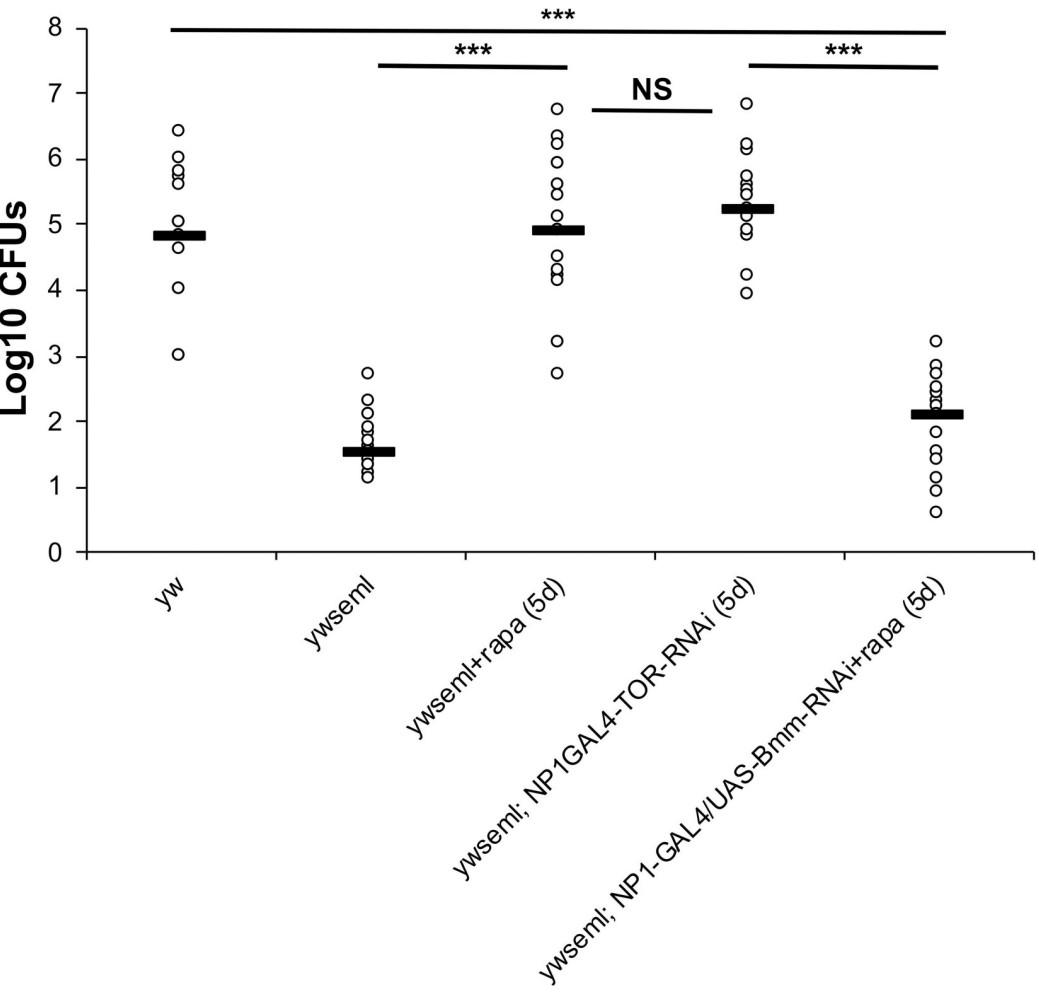

**Fig 7. Reduction of bacterial density is connected to gut lipid catabolism.** Rapamycin treatment or TOR-RNAi in ECs of *ywseml* flies restored intestinal bacterial CFUs at the levels of the genetic background of the levels of the genetic background of *yw*. However, this was not the case when the lipase *bmm* was knocked down in ECs. N = 15/genotype/treatment a total of three independent experiments (each with n = 5/genotype/treatment). Values of mutants and controls were statistically compared using student's t-test (***p<0.001, NS is non-significant).

background, suggesting that loss of Toll signalling in *PGRP-SA*<sup>seml</sup> flies resulted in a TOR-mediated suppression of lipid catabolism ([Fig 6]). Symmetrically, inhibition of TOR through RNAi in enterocytes was also able to put a brake on fat accumulation in the gut. This was dependent on 4E-BP. Indeed, treatment of *PGRP-SA*<sup>seml</sup>*; NP1*<sup>ts</sup>*>4E-BP*<sup>RNAi</sup> flies with rapamycin was ineffective in restoring normal triglyceride levels, indicating that 4E-BP was needed in enterocytes to regulate intestinal fat levels ([Fig 6]). As differences in food intake can modulate triglyceride reserves, we measured food intake during a week of observation (in 1 to 7-day old mated female flies) using a capillary feeding assay [CAFE assay, 46]. Food intake was statistically indistinguishable between flies deficient for *PGRP-SA*<sup>seml</sup> (treated or untreated with Rapamycin), *ywPGRP-SA*<sup>seml</sup>*; NP1>mTOR*<sup>RNAi</sup> and *PGRP-SA*<sup>seml</sup>*; NP1*<sup>ts</sup>*>4E-BP*<sup>RNAi</sup> (treated with Rapamycin) compared to *yw* controls ([S13 Fig]).

### Lipid catabolism in the gut is connected to the density of cultivable bacteria

To test if the restriction in intestinal lipid catabolism (and thus accumulation of fat in the gut) was the cause of the drop in cultivable bacterial density, we measured the influence of the loss

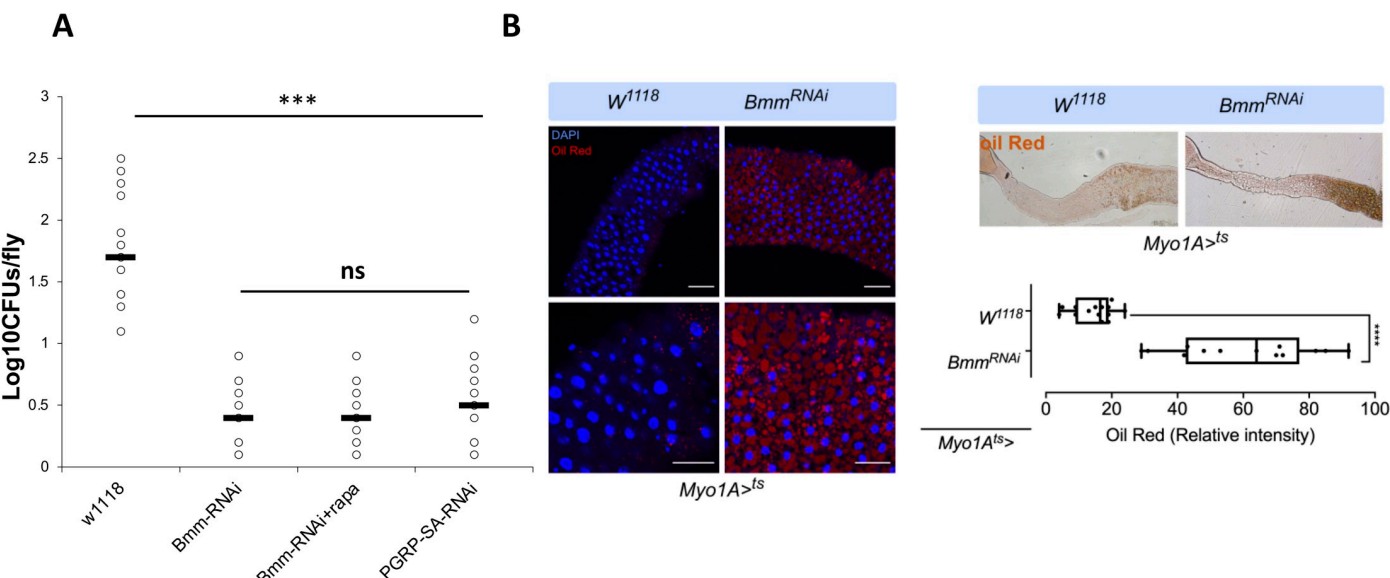

**Fig 8. RNAi of the Bmm lipase increases intestinal lipid accumulation and reduces bacterial density. (A)** Silencing of *bmm* expression in enterocytes (through *Myo1A^ts*-GAL4) resulted in significant reduction of enteric CFUs. **(B)** This was coupled to an accumulation of lipids in the gut as stained and quantified with Oil Red. Statistical comparisons were conducted using student's t-test (ns = not significant, ***p<0.001, ****p<0.0001).

of the lipid storage droplet-associated TAG lipase Brummer (*bmm*), a homolog of human adipocyte triglyceride lipase [47]. As observed previously, *PGRP-SA^seml* mutants had a decreased density of cultivable bacteria, while *PGRP-SA^seml* flies treated with rapamycin or with TOR RNAi (in enterocytes) displayed levels of cultivable bacteria comparable to the *yw* background (Fig 7). However, this was not the case when rapamycin treatment was accompanied by *bmm* knock down in enterocytes, indicating that Bmm activity was important in restoring the density of cultivable gut bacteria (Fig 7). Silencing *bmm* in enterocytes of wild type flies significantly decreased intestinal CFUs (Fig 8A). Moreover, *bmm* silencing in enterocytes significantly increased lipid accumulation in the gut as seen and quantified with Oil red (Fig 8B).

## Intestinal bacterial sensing, bacterial density, and lipid catabolism

To ascertain whether the presence of bacteria and their sensing by PGRP-SA were important factors in connecting the maintenance of intestinal bacteriome with lipid catabolism, we stained axenic flies with Oil Red. Axenic wild type flies had an elevated level of gut lipids in comparison to conventionally reared ones, but this clear trend was at the limit of statistical significance (p = 0.15) (Fig 9). To check for the necessity of binding *in vivo*, we generated three UAS-PGRP-SA transgene mutants. These were point mutations in Serine 101, Tyrosine 126, and Serine 184 to Alanine (Fig 10A). None of the residues chosen for mutagenesis is thought to be involved in extensive packing interactions. Hence, alterations of these residues are not expected to disrupt the tertiary structure of PGRP-SA [48].

S101 sits at the base of the peptidoglycan (PG)-binding groove (Fig 10A) and the S101A mutant increases binding to bacterial peptidoglycan *in vitro* [48], suggesting that the removal of the hydroxyl group of S101 may create a better binding surface for PG. In contrast, Y126A and S184A abolish PG binding *in vitro* [48]. Expressed in enterocytes through an UAS, a wild type PGRP-SA transgene was able to rescue the significant reduction of intestinal CFUs in

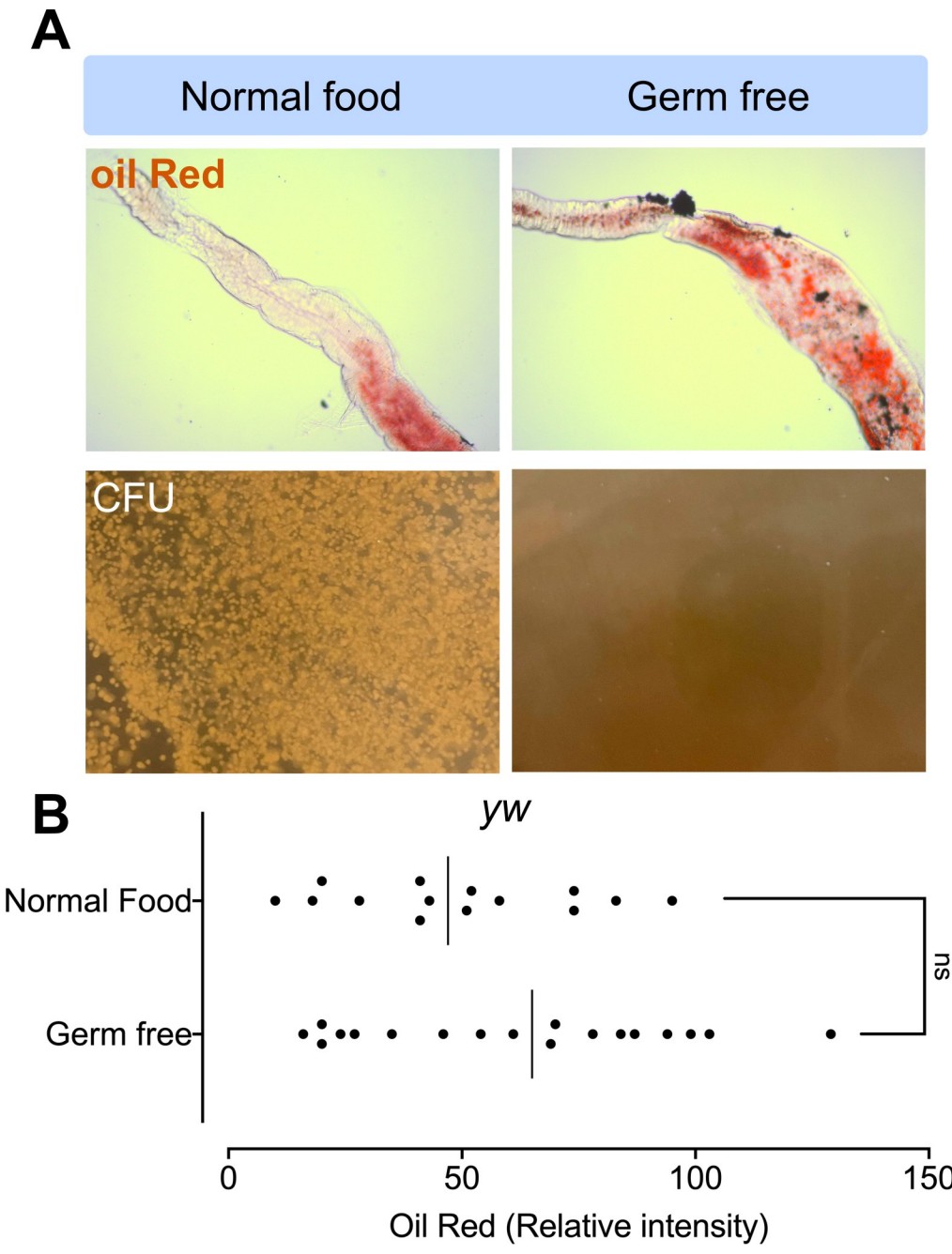

**Fig 9. Axenic flies show a modest accumulation of enteric lipid levels. (A)** Flies cultured in axenic conditions had on average more Oil red-stained guts. **(B)** Despite the observed trend this was at the margin of statistical significance (p = 0.15) as calculated using student's t-test (ns = non-significant).

5-day old *PGRP-SA^seml^* flies (Fig 10B). *UAS-PGRP-SA^S101A^*, was also able to rescue the reduction in gut bacterial density (Fig 10B). In contrast, expression of *UAS-PGRP-SA^Y126A^* or *UAS-PGRP-SA^S184A^* in enterocytes were unable to rescue enteric CFUs (Fig 10C). This indicated that only a PGRP-SA protein with an intact ability to bind PG was able to rescue the loss of gut bacterial density.

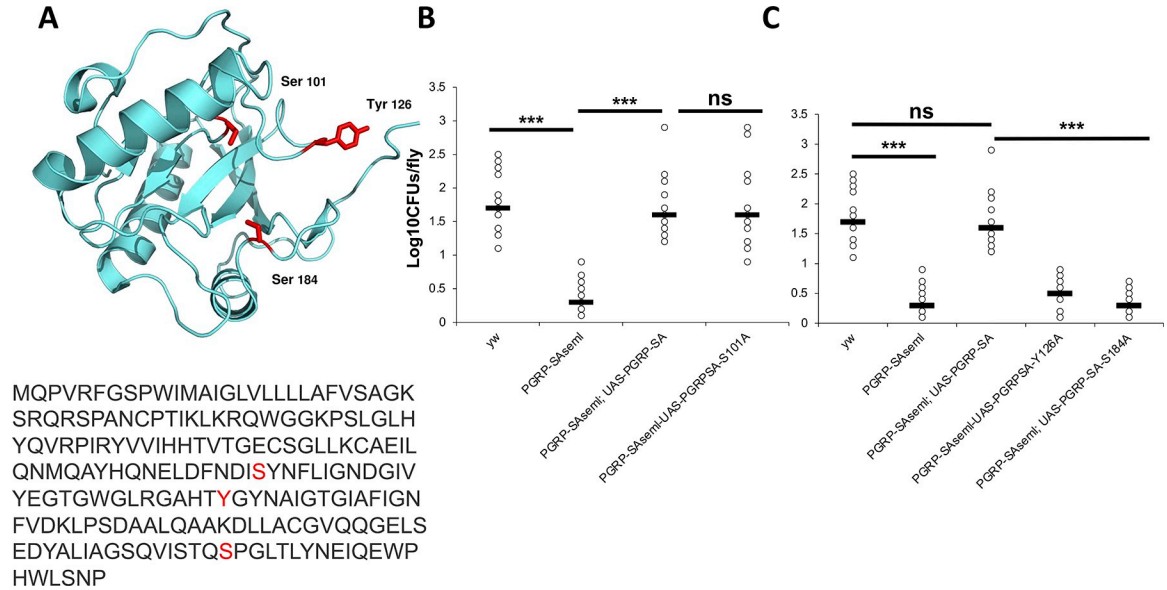

MQPVRFGSPWIMAIGLVLLLLAFVSAGK
SRQRSPANCPTIKLKRQWGGKPSLGLH
YQVRPIRYVVIHHTVTGECSGLLKCAEIL
QNMQAYHQNELDFNDI**S**YNFLIGNDGIV
YEGTGWGLRGAHT**Y**GYNAIGTGIAFIGN
FVDKLPSDAALQAAKDLLACGVQQGELS
EDYALIAGSQVISTQ**S**PGLTLYNEIQEWP
HWLSNP

**Fig 10. Bacterial sensing by PGRP-SA is important for the maintenance of intestinal bacterial density. (A)** Mutations (to Alanine) were introduced in three residues spanning the peptidoglycan binding groove. *In vitro* studies [48] have indicated that S101A increases peptidoglycan (PG) binding, while Y126A and S184A abolish PG binding. **(B)** Both a wild-type copy of PGRP-SA and PGRP-SA$^{S101A}$ were able to rescue the significant reduction of CFUs caused by the loss of function PGRP-SA$^{seml}$. **(C)** In contrast, PGRP-SA$^{Y126A}$ and PGRP-SA$^{S184A}$ were unable to rescue loss of gut bacterial density. Statistical comparisons were conducted using student's t-test (ns = not significant, $^{***}$p<0.001).

## Discussion

In the absence of the bacterial receptor PGRP-SA from enterocytes, we observed a reduction in intestinal bacterial density. It was restored with the use of rapamycin, which targets TORC1 or by knocking-down TOR in enterocytes. This suggested that loss of the immune receptor PGRP-SA, generated a metabolic environment unfavourable for intestinal bacterial growth. Our results indicated that at the centre of this relationship was 4E-BP, which is activated by Toll and suppressed by TORC1. In keeping with this, *PGRP-SA$^{seml}$; NP1GAL4>4E-BP$^{RNAi}$* flies treated with rapamycin were unable to restore gut bacterial density. Intestinal lipid catabolism downstream of 4EBP was paramount for the maintenance of cultivable bacterial density because the loss of the lipase Bmm blocked restoration of gut bacteria after rapamycin treatment. Silencing of *bmm* in enterocytes caused intestinal lipid accumulation and prevented any restoration via rapamycin in *PGRP-SA$^{seml}$* flies.

Our results indicate that downstream of Toll, intestinal triglyceride levels were under 4E-BP control in enterocytes. Although the phenomenon of cultivable bacteriome reduction in *PGRP-SA$^{seml}$* flies was readily manifested in larvae and young flies, our results indicated that it was also there in older flies. Conventional fly rearing techniques ensure a steady stream of defaecation and re-introduction of bacteria over time. However, when food vials were changed rapidly re-infection was reduced and CFUs in 30-day old flies were significantly lower in *PGRP-SA$^{seml}$* than *yw* flies. Preservation of the bacteriome was dependent on PG recognition as the rescue of enteric CFUs in *PGRP-SA$^{seml}$* flies was only possible with PGRP-SA transgenes that had an intact PG binding ability. This indicated that bacterial *sensing* was the initial trigger point to activate the process.

Our working model is depicted in Fig 11. PGRP-SA recognises components of the intestinal bacteriome and activates the Toll pathway in enterocytes. In turn, this keeps increased 4E-BP

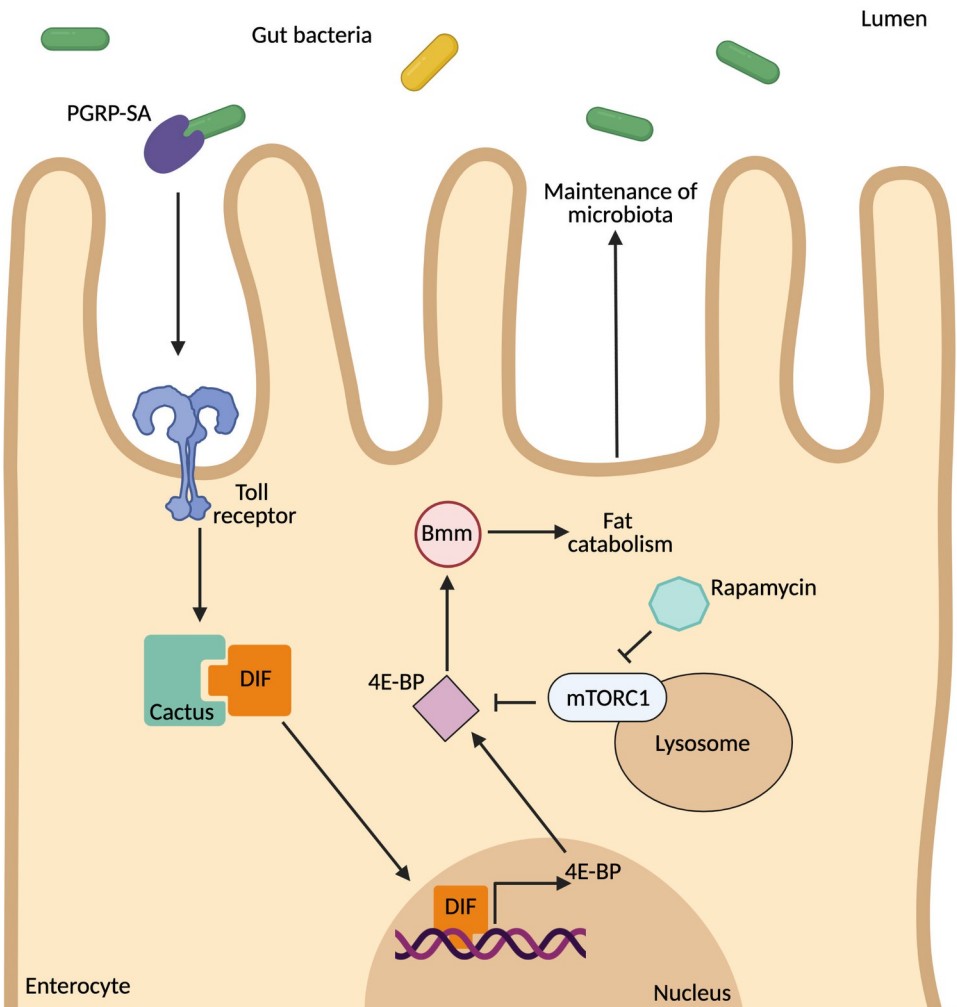

**Fig 11. A schematic model outlining the role of PGRP-SA/Toll/Dif in the retention of the gut bacteriome.**
PGRP-SA recognises components of the intestinal bacteriome and activates the Toll pathway in enterocytes. This
increases 4E-BP transcription/4E-BP protein phosphorylation in enterocytes. 4EBP is important for maintaining
normal density of commensal bacteria. We hypothesise that Bmm-mediated lipid catabolism is regulated by 4E-BP and
released triglycerides act as fuel for the maintenance of commensal bacteria.

transcription/4E-BP protein phosphorylation in enterocytes, preserving a steady rate of intestinal lipid catabolism. The latter is important for maintaining normal density of commensal bacteria. We hypothesise that Bmm-mediated lipid catabolism is regulated by 4E-BP and released triglycerides act as fuel for the maintenance of commensal bacteria. In keeping with this, stopping lipid catabolism by silencing the Bmm lipase in ECs resulted in accumulation of lipids and reduction of enteric CFUs. According to the model, bacteria should trigger lipid catabolism and 5-day old axenic flies showed a clear trend for lipid accumulation in their gut, but this was marginally not statistically significant. Studies with *Vibrio cholera*, have shown that intestinal acetate leads to deactivation of host insulin signalling and lipid accumulation in enterocytes, resulting in host lethality [49]. Loss of PGRP-SA/Dif leads to a decrease in lifespan. Whether this is due to the long-term accumulation of lipids is an open question.

More work is needed to understand whether/how stored intestinal lipids maybe released to circulation, how commensal bacteria receive them and which component(s) of the bacteriome are recognised by PGRP-SA.

## Material and methods

### Drosophila stocks and procedures

Yellow white (*yw*) *Drosophila* strain *yw*$^{67c23}$, *w*$^{1118}$, and DGRP (Drosophila Genetic Reference Panel) 25174 were used as control genetic backgrounds to compare with mutants *PGRP-SA*$^{seml}$ [34] and *dif*$^{1}$ [27] or other flies with cell-specific knockdowns using the UAS-GAL4 system. The *Drosophila* strains used for the latter were UAS lines from the Vienna Stock centre namely *UAS-PGRP-SA*$^{RNAi}$ (GD line 37470) and *UAS-4E-BP*$^{RNAi}$ (KK line 100739), NP1-GAL4/Cyo; drsGFP/TM3 (RRID:BDSC 67088) as well as UAS-TOR$^{RNAi}$ (A15495 TRiP line) from Bloomington Stock Centre. Lines UAS-PGRP-SA, UAS-PGRP-SA$^{S101A}$, UAS-PGRP-SA$^{Y126A}$ and UAS-PGRP-SA$^{S184A}$ were constructed for this study. For assimilating the *PGRP-SA*$^{seml}$ and *dif*$^{1}$ mutations in the 25174 genetic background, mutant female flies were crossed to 25174 males with balancers on the first (FM7) and second chromosome (CyO) and individual lines (one founder female X one 25174 male with balancers) were established. Female progeny from these were then backcrossed with 25174 (without balancers) for another 3 generations. At the end of the 3 generations, 20 individual crosses (one 25174 male with balancers X one female from each line) were set up, balanced F1 was crossed and non-balanced F2 progeny were screened for the presence of the relevant mutation in homozygosity.

During all experiments flies were reared at 25˚C, 70% humidity, on cornmeal-molasses medium under a 12-hour light: dark cycle. Backup stocks for each line were maintained at 18˚C. Experimental flies were allowed to mate for two days after collection in vials with fresh food. They were then segregated based on sex into separate vials. Around 15 to 20 flies of the same sex were housed in each vial for every experiment and tipped into fresh food every two days till 30 days. Rapamycin (37094, Sigma-Aldrich, USA) stock (10mg/ml) was freshly prepared in 100% ethanol and added into microwaved fly food to a final concentration of 200μM. The same amount of ethanol (1:50 dilution) was also added to fly food as a vehicle control. Flies were fed with Rapamycin food for three days after which their guts were isolated for different experiments. The fly food recipe was (to 30L): Maize Flour 1.8kg, Malt Extract 1.8kg, Molasses 500g, Soya Flour 216g, Yeast Powder 366g, Agar (Fisher Scientific, BPE2641-1) 169g, Methyl 4-hydroxybenzoate (Nipagin, Sigma-Aldrich W271004) 74g, Ethanol (VWR) 703ml, [95% propionic acid (Sigma-Aldrich P5561) 5% phosphoric acid (Sigma-Aldrich P5811) solution 140ml], Water 28L.

### Culture-dependent quantification of gut microbiota

Individual midguts were dissected in Phosphate-buffered saline (PBS) solution and then homogenized in MRS broth (CM0359, Thermo Fisher) using a sterile needle. The resulting suspension was then plated on MRS agar (CM0361, Thermo Fisher), and incubated at 30˚C for 48 hours. The colonies in each plate were counted, and log10 values of the colony forming units (CFU's) were calculated. At least fifteen individual guts of any individual fly line were plated on individual MRS plates following the steps above. The log10 CFU values of mutants and controls were statistically compared using student's t-test, and graphs were plotted in GraphPad Prism v.8.0.

### Culture independent quantification of gut microbiota

Identification of microbial families is widely performed by analysing their 16s ribosomal RNA sequences using the hypervariable region V3 to identify microbial composition of the *Drosophila* gut.

*DNA isolation and processing* Forty midguts were isolated from 5 day and 30-day old flies of each strain in PBS, and their genomic DNA (gDNA) was extracted using QIAamp DNA Mini

Kit (51304, Qiagen, Germany) according to manufacturer's instructions. A no gut sequencing control was also included to rule out any contaminant bacteria in the reagents. Readout of the composition of Drosophila microbiome is challenged by the intra-cellular endosymbiont Wolbachia, which is known to infect *Drosophila* [50]. To remove/reduce the 16S reads of Wolbachia, the extracted DNA was digested with the restriction enzyme BstZ17I (R3594S, NEB, UK) which cuts at a unique restriction site in the 16S region of Wolbachia RNA only [50]. Approximately 500ng of genomic DNA (gDNA) from each fly line was incubated with BstZ17I for 2 hours, followed by heat inactivation of the enzyme. The digested gDNA was then used as a template to amplify the 16S hypervariable regions V1-V3. This amplification step was carried out with Phusion HF DNA polymerase (MO530L, NEB, UK) by PCR. Wolbachia 16S region are not amplified during this step, due to the aforementioned digestion, resulting in a higher yield of amplicons of the gut microbiota. Digestion of the V1-V3 region of Wolbachia after PCR amplification of its 16S gDNA resulted in the production of a 200bp and 300bp bands, both of which were confirmed to belong to Wolbachia through sequencing. The Qiagen PCR purification kit (27106, Qiagen, Germany) was used to purify the PCR products, which was then used as a template to amplify V3 hypervariable region (180-200bp). After amplification, the resulting V3 product was run on a 1.2% agarose gel, and a 200bp V3 band was excised and purified using the QIAquick Gel Extraction Kit (28706, Qiagen, Germany).

*V3 library preparation* Up to 100ng of V3 DNA samples (measured using Picogreen assay (P11496, Thermo Fisher, USA) according to manufacturer's instructions) were used for library preparation. The NEB NextFast DNA library Prep Set for Ion Torrent kit (E6270, NEB, UK) was used to prepare DNA libraries. The NEB end repair enzyme from the kit was utilized to remove DNA overhangs and ensure that each DNA molecule contains a free 5' phosphate and 3' hydroxyl groups for adaptor ligation. This was carried out using the reaction mix and thermocycling conditions according to manufacturer's instructions. Unique Ion Xpress Barcode Adaptors (4471250, ThermoFisher Scientific) along with universal adaptors supplied in the kit were then ligated to the ends of the DNA strands and amplified according to manufacturer's instructions. The resultant DNA libraries were then purified using Agencourt AMPure XP DNA purification beads (A63881, Beckman Coulter, Germany) to remove excess barcodes from the product. Samples were run on 2% E-gel (G661012, ThermoFisher Scientific, USA) using the SizeSelect 2% setting (E-Gel iBase Invitrogen) and 300bp libraries were extracted and purified twice using Agencourt AMPure XP DNA purification beads (A63881, Beckman Coulter, Germany). The purified libraries were then amplified using 6–8 rounds of PCR and quantified with the KAPA library quantification kit for Ion Torrent (KK4812, KAPA Biosystems, USA) using specific primers against Ion Torrent DNA standards (KK4812, KAPA Biosystems, USA). Samples were then mixed equally to a final concentration of 350pM, loaded onto the Ion 314TM Chip V2 using Ion Chef system, and sequenced using Ion Proton system, (ThermoFisher Scientific) according to manufacturer's instructions.

The Ion Proton system works under the principle of sequencing by synthesis. In this method, identification of nucleotide bases is performed by tracking changes in the pH. Formation of a phosphodiester bond during dNTP polymerization leads to the release of protons, which can be detected as a pH change. The Ion Torrent system detects these changes in the pH using ISFET (Ion sensitive field effect transistor) detectors placed in each micro-well. Signals from millions of ISFET detectors are relayed through CMOS array chips, which presents the nucleotide readout [51].

*Sequence analysis* Sequencing data produced by the Ion Proton system was processed using the ThermoFisher Scientific Ion Reporter software. Sequences were uploaded as BAM files and analysed using a standard designed metagenomics 16S workflow version 5.10 (Thermo Fisher) with V3 primers mentioned above and customized parameters as described later. The

sequences were referenced against Curated MicroSEQ(R) 16S Reference Library v2013.1. Reads were filtered so that single end primers were detected in every read, sequences less than 115 bp were excluded from the results, and all reads that would align with at least 90% of the database sequence were included. Operational taxonomic Unit (OTU) with at least one unique read was picked up. Cut offs at 97% and 99% sequence identity were used for assignment at genus and species levels respectively. The difference between two consecutive hits was set at 0.8%. Reads mapping to the sequences of the *Drosophila* symbionts belonging to *Anaplasmataceae* (which do not constitute members of the gut microbiome), were excluded from the downstream analysis. The relative abundance of each family was estimated by dividing mapped reads from each family by the total mapped reads (minus *Anaplasmataceae*) per sample. The diversity of microbiota within and between each sample was measured by plotting alpha (Simpson's 1-D, and Shannon H) and beta diversity (PCA plot) indices respectively using PAST3 software. Alpha diversity takes into account both species richness and abundance within the same study. Simpson's and Shannon's indices are commonly used to score the alpha diversity of a population. Simpson's index (D) measures the probability of selecting two individuals from a population belonging to the same species, while Shannon's diversity index (H) measures the species richness and evenness in terms of uncertainty of predicting species in a sample (ie. entropy) [52]. Beta diversity analysis compares the bacterial composition between sample populations and plots a representative Principle component analysis (PCA) plot. Distance between points in the PCA plots represents the extent of similarity/dissimilarity between samples based on their composition [53]. All graphs in these experiments were produced in GraphPad Prism v.8.0. PCA plots were prepared in the PAST3 software and edited using Inkscape v.0.92.4.

## Measuring gene expression with qPCR

Ten flies from each line were dissected in PBS and their guts were isolated. Intestines were homogenized, and total RNA was extracted using Norgen total RNA purification plus kit (48400, Norgen-Biotek, Canada). The RNA concentration (50–500 ng/μl) and purity (260nm/280nm absorbance ratio) was measured using Nanodrop. The absorbance ratio of RNA (260nm) and proteins (280nm) indicates the purity of RNA in the sample. For one single fly line, RNA was isolated from a pool of 10 guts. This was performed three times per fly line. Total RNA was used as a template to prepare cDNA using the SuperScript VILO cDNA Synthesis Kit (11754050, Thermo Fisher, USA) based on manufacturer's instructions. QPCR was performed using cDNA produced from total RNA (described above) of fly guts as template. The SensiFAST SYBR No-ROX Kit (BIO-98020, Bioline, Germany) and sequence specific primers were used to quantify transcript levels of 4E-BP against the housekeeping gene RP49 with the help of the Qiagen Rotor gene Q qPCR instrument. The amount of cDNA added was standardized to 25ng/μl of the precursor RNA. Quantification was performed according to manufacturer's instructions. The ribosomal protein 49 gene (RP49) was used it as an internal control. Three biological repeats were each loaded in triplicates (three technical repeats) (n = 9) and ΔΔCT values were plotted and analysed for statistical significance using student's t-test in GraphPad Prism v.8.0.

## Lifespan analysis

Prior to carrying out lifespan assay, *yw* and *yw dif^1* flies were backcrossed for two generations to remove any inbred mutations. Two-day old mated female flies were collected in batches of 10 per vial in 12 vials (120 per fly line). These flies were tipped into fresh food every 2 days until all the flies died. Survival was monitored by counting the number of dead flies every

alternate day. Unaccounted flies that flew away during tipping or died because of manual handling were recorded as censored. The number of dead and censored flies were tabulated in a custom excel template designed by Matthew Piper (Monash University, Australia) which was used to produce the lifespan graph. Lifespan of two different fly strains were statistically compared using Log-rank test in which the difference between two populations at a time point is examined by analysing the probability of an event (death) to occur [54].

### Triglyceride measurement

Newly hatched L1 larvae were seeded in vials as at density of 50/vial and grown at 25˚C to synchronise the culture. Adults at the appropriate age were processed in batches of eight for males and six for females. Only male data are presented (of note that females did not deviate from the results obtained). Samples were processed immediately in homogenisation buffer [0.05% Tween-20 and 2x protease inhibitor (Roche) in $H_2O$]. After centrifugation (5000 rpm, 1min) the supernatant was transferred to a new tube and span again (14000 rpm, 3mins at 4˚C). To measure triglycerides 80μl of the above supernatant were mixed with 1ml of the Triglyceride Reagent (Thermo-Fisher) and incubated for 10mins at 37˚C. Measurements were taken at $OD_{520}$ and compared with a standardization curve. To measure protein levels, 100μl of the final supernatant was combined with 700μl of $H_2O$ and 200μl of Bio-Rad Protein Assay Reagent and incubated at room temperature for 3mins. Measurements at $OD_{595}$ were compared with a standardization curve.

### Gut dissection and immunostaining

For gut imaging, guts from anesthetized flies were dissected in Schneider's medium and fixed for 30 min in 4% paraformaldehyde (in PBS), rinsed in PBS and then three times washed (5 min each) in wash solution, 0.1% Triton X-100 (Sigma-Aldrich) in PBS. The tissue was blocked for 60 min in blocking solution (0.1% Triton X- 100, 2% BSA (Sigma-Aldrich) in PBS and immunostained with primary antibodies overnight at 4˚C. Samples were then washed 4 x 5 min at room temperature (RT) In wash solution, incubated with secondary antibodies at RT for 2 hours, washed again as before and were them stained with DAPI 1:1000 (Sigma-Aldrich). Washed guts were mounted in slides with vectorshield mounting media (Vector Laboratories). The following primary antibodies were used: mouse anti-β- galactosidase (40-1a-S, Developmental Studies Hybridoma Bank, Iowa, USA) - 1:1000; goat anti-HRP (123-165-021, Jackson ImmunoResearch Labs. Inc.)-1:500. we used donkey anti-mouse Alexa 568 antibody (Invitrogen) - 1:250 and donkey anti- goat Alexa 568 antibody (Invitrogen) - 1:250.

### Imaging data analysis

Guts were imaged at 20x magnification, and all GFP marked cells (esg > Gal4) co-localised with DAPI small nuclei were counted in an area of approximate size that extended anteriorly from Boundary 2–3; plotted values are the number of GFP marked cells per unit area or per total number of DAPI stained cells. Images were analysed using ImageJ software.

### Capillary feeding assay (CAFE assay)

Food intake was analysed as previously described [46] with some modifications. 50 flies per genotype were tested. Batches of 10 flies were placed in vials with wet tissue paper at the bottom and a capillary food source containing a blue dye. Feeding was monitored for 8 hours (light ON) and 1 hour (light OFF). Feeding amount was recorded every 1 hour and the capillaries were replaced every 2 days.

## Statistical analysis

Data was analysed using GraphPad Prism 6 or R. First a D' Agostino and Pearson omnibus Normality test was conducted. If the data was found to fit a normal distribution, parametric tests were used, first ANNOVA and then a Tukey's multiple comparisons test. For the Thor-LacZ count data in the cases that did not fit the normal distribution we conduct Kruskal-Wallis test for the followed by the Dunn's multiple comparisons test to clarify the significance. R was used to analyse the GFP count data, it was fitted to a generalised linear model using a quasi-Poisson regression and then ANNOVA and Tukey's multiple comparisons tests were employed to look for significance. For qPCR gene expression data was standardized by series of sequential corrections, including log transformation, mean centring, and autoscaling [55].

## Oil red staining

Oil red staining was performed as previously described [56]. Briefly, fly guts were dissected in phosphate buffered saline (PBS) on ice and fixed with 4% formaldehyde for 45 min. Guts were washed with consecutive applications of PBS, double-distilled water, and a 60% isopropanol solution. A previously prepared solution of 0.1% Oil Red O (Sigma-Aldrich) stock in isopropanol was used to make a fresh working stock of 6:4 dilution in water. Guts were incubated for 20min and then washed with 60% isopropanol and water (thrice).

## Antibody staining

p4EBP and PH3 staining was done as follows. Adult flies were anesthetized and their midgut was dissected in 1X PBS. Guts were placed onto poly-lysine slides and were fixed with 4% formaldehylde (28908, VWR diluted in PBS) for 30minutes. The fix was discarded and the guts were then washed with PBST (1XPBS + 0.1% Triton X-100 (T8787-250 mL, Sigma, Darmstadt, Germany)). The samples were then blocked with PBST + 1% BSA (1062, Gerbu, Heidelberg, Germany). After blocking, the guts were incubated overnight with the primary antibody at 4 degree Celcius. Guts were thoroughly washed with PBST and subsequently incubated at RT in fluorescently labelled secondary antibody (final concentration 1:3000) for 1.5hrs. Stained midguts were then mounted in Vectashield containing DAPI (H-1200, Linaris, Burlingame, CA, USA). The control samples and experimental samples were stained on the same slide for direct comparison. Primary antibodies used were rabbit anti -4EBP (Thr37/46, Cell Signaling, #2855, 1:200), and rabbit anti-pH3 (Cell signalling, #9701L, 1:500), and secondary antibody used was chicken anti rabbit AF594 (1:3000, A21442, Invitrogen,Waltham, MA, USA).

## Supporting information

**S1 Fig. *PGRP-SA^seml^* mutant flies have less intestinal progenitor cells. (A)** In the absence of infection, ISC (HRP positive, GFP positive) divide to produce EBs (HRP negative, GFP positive). However, in 20-day old flies that were deficient for PGRP-SA this division was not observed (see also insets). **(B)** Quantification of progenitor cells and ISCs showed that these were significantly reduced (*p<0.05; error bars display 95% confidence intervals, guts from 4 biological repeats were analysed). GFP expression was directed by the UAS dependent mCD8GFP transgene, which marked the cell membranes of the progenitor cells including ISCs and EBs (DAPI all nuclei).
(TIFF)

**S2 Fig. Toll10B rescues progenitor cell number in the absence of *PGRP-SA*.** Expressing a UAS- transgene of a gain of function version of the Toll receptor (Toll10B) in progenitor cells activated ISC division in the absence of functional PGRP-SA, as indicated by staining midguts

of 5-day old females with an anti-phospho-histone-3 antibody (PH3+) (left panel) and quantified (right panel). Of note, that ISCs are the only PH3+ cells of the intestinal epithelium.
(JPG)

**S3 Fig. Loss of Dif but not Relish results in loss of bacterial density. (A)** Dif RNAi in enterocytes resulted in a significant reduction in bacterial CFUs. **(B)** In contrast, loss of Relish resulted in the significant increase of two major components of the intestinal bacteriome namely, *Lactobacillus* and *Acetobacter* spp. Statistical comparisons were conducted using student's t-test (*p<0.1, **p<0.01, ***p<0.001).
(JPG)

**S4 Fig. Members of the Toll signalling pathway are involved in maintaining the intestinal bacteriome. (A)** *PGRP-SA$^{seml}$* flies showed a significant reduction in lifespan. Sex-specific pairwise statistical comparisons were conducted using the log-rank test (***p<0.001). **(B)** *spz$^{rm7}$* flies showed significantly reduced intestinal CFUs. Statistical comparisons were conducted using student's t-test (ns = not significant, ***p<0.001).
(JPG)

**S5 Fig. 30-day old *PGRP-SA$^{seml}$* flies lose their bacterial density faster than their genetic background when food is changed every 12h. (A)** Loss of bacterial density in *yw* flies was stabilised at day 20 (from day 1 of adulthood) whereas **(B)** similar treatment of in *PGRP-SA$^{seml}$* flies resulted in loss of bacterial density beyond 20-days as **(C)** was seen in a direct comparison of 30-day old *yw* and *PGRP-SA$^{seml}$* flies. Values of mutants and controls were statistically compared using student's t-test (***p<0.001).
(TIFF)

**S6 Fig. Loss of Toll signalling results in loss of intestinal bacteria. (A)** Semi-quantitative PCR of 16S rRNA (right gel) from isolated guts showed an age-dependent reduction in total intestinal bacteria when comparing *yw* to *PGRP-SA$^{seml}$* or *dif$^1$* to the internal control (actin; left gel). **(B)** Quantification of 16S bands (using Image J) relative to actin showed a significant reduction in the quantities of 16S (***p<0.001, ns = non-significant, by student's t-test). Every dot represents the quantification of one band relative to the respective actin control (n = 10).
(JPG)

**S7 Fig. Repeat experiment to examine the intestinal bacteriome diversity of *yw* and *PGRP-SA$^{seml}$* flies (see [Fig 2]). (A)** The graph represents the relative abundance of bacterial families observed in the gut of 5-day and 30-day old female *yw*, *ywseml* flies revealed by 16S next-generation sequencing. The x axis represents y strains of different ages, and the y axis represents relative mapped reads. (n = 40 guts/strain). **(B)** PCA plot to show that samples of the same genotype (in circles) were statistically indistinguishable whereas between genotypes were significantly different (***p<0.001).
(TIFF)

**S8 Fig. Rapamycin restores intestinal progenitor cells (IPCs) in *PGRP-SA$^{seml}$* mutants.** In the anterior midgut (see schematic), IPCs were reduced in *PGRP-SA$^{seml}$* mutants. Addition of rapamycin was able to restore them. This is a representative result from 15 guts.
(TIFF)

**S9 Fig. Effect of Rapamycin treatment and gut specific knockdown of mTOR on the microbial composition of 30 day old. (A).** The graph represents the relative abundance of microbial families observed in the gut of 30-day old flies *yw*, *ywseml* and flies treated with rapamycin and *ywseml; NP1>mTOR$^{RNAi}$* (+/- rapamycin) as revealed by 16S next generation sequencing. The

x axis represents y strains of different ages, and the y axis represents relative mapped reads. (n = 40 guts/strain). **(B)** Principal component analysis (PCA) of the above 30-day old flies (n = 40 guts/strain). **(C)** Principal component analysis (PCA) comparing of 5 and 30-day old of the above genotypes (for 5-day old flies see **Fig 4**). (n = 40 guts/strain).
(TIFF)

**S10 Fig. Systemic *C. albicans* infection activates transcription of the *Drosophila* 4E-BP (Thor) in enterocytes. (A)** *thor-lacZ; esg-ts<GFP* flies injected with *C. albicans* were sampled 36 hours post-infection and compared to non-treated (homeostasis) or those injected with PBS (sterile injury). Gut cells stained with DAPI (blue), anti-β-galactosidase (red) and anti-GFP expressing cells (marking both ISCs and EBs). Shown are representative images from the anterior midgut taken at 63x. **(B).** Quantification of *thor-lacZ* expression upon systemic infection. Intensity measured using ImageJ, subtraction of the background was performed for all samples. Ten guts were analysed (approximately 50 cells analysed per gut), 95% confidence intervals displayed, *p<0.05, *** p<0.001.
(TIF)

**S11 Fig. Overexpression of Toll10B increases the phosphorylated form of 4E-BP.** Overexpression of the constitutive form of the Toll receptor, Toll10B, in progenitor cells (GFP-labelled in Toll10B panels) caused a cell-autonomous increase in the occurrence of the phosphorylated form of 4E-BP (p4EBP, arrows) even in the absence of functional PGRP-SA (lower panel). *Myo1Ats>UAS-Rheb* was used as a positive control for p4EBP staining in enterocytes (GFP labelled in *UAS-Rheb* panels). Bar is 30 μm.
(JPG)

**S12 Fig. 4EBP is important for the maintenance of cultivable gut bacterial density.** Depletion of 4EBP via RNAi in enterocytes, reduced CFUs in both males and females when compared to the *NP1-GAL4* driver (****p<0.0001, **p<0.01 student's t-test).
(TIFF)

**S13 Fig. Food intake is not influenced by lack of PGRP-SA, 4E-BP, TOR or administration of rapamycin.** Food consumption was measured by the CAFE method in mated females from day 1 to day 5 of adulthood; each dot represents a vial (of 10 flies each) (n = 15 vials for each genotype and treatment); no comparison was statistically significant (p>0.05, unpaired t-test).
(TIFF)

## Author Contributions

**Conceptualization:** Marcus Glittenberg, Petros Ligoxygakis.

**Formal analysis:** Shivohum Bahuguna, Magda Atilano, Marcus Glittenberg, Jun Zhou, Petros Ligoxygakis.

**Funding acquisition:** Michael Boutros, Petros Ligoxygakis.

**Investigation:** Shivohum Bahuguna, Magda Atilano, Marcus Glittenberg, Dohun Lee, Srishti Arora, Jun Zhou, Siamak Redhai, Petros Ligoxygakis.

**Methodology:** Shivohum Bahuguna, Magda Atilano, Marcus Glittenberg, Petros Ligoxygakis.

**Resources:** Lihui Wang, Jun Zhou, Siamak Redhai, Michael Boutros.

**Supervision:** Michael Boutros, Petros Ligoxygakis.

**Writing – original draft:** Petros Ligoxygakis.

**Writing – review & editing:** Shivohum Bahuguna, Petros Ligoxygakis.

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
