## [Decision Letter · Decision Letter 0]

2 Jun 2021

Dear Dr Ligoxygakis,

Thank you very much for submitting your Research Article entitled 'Bacterial recognition and Toll signalling sustain commensal gut bacteria in Drosophila.' to PLOS Genetics.

The manuscript was fully evaluated at the editorial level and by independent peer reviewers. The reviewers appreciated the attention to an important problem, but raised some substantial concerns about the current manuscript. Based on the reviews, we will not be able to accept this version of the manuscript, but we would be willing to review a much-revised version. We cannot, of course, promise publication at that time.

If you decide to revise the manuscript for further consideration at PLOS Genetics, please aim to resubmit within the next 60 days, unless it will take extra time to address the concerns of the reviewers, in which case we would appreciate an expected resubmission date by email to plosgenetics@plos.org.

[LINK]

We are sorry that we cannot be more positive about your manuscript at this stage. Please do not hesitate to contact us if you have any concerns or questions.

Yours sincerely,

Gregory P. Copenhaver, Ph.D.

Editor-in-Chief

PLOS Genetics

Reviewer's Responses to Questions

**Comments to the Authors:**

Reviewer #1: This manuscript from Bahuguna and co-workers is an interesting exploration of the role of the Toll pathway in the interaction between Drosophila and its gut microbiota. The topic is of interest and the work appears sound, but there are a few points where the manuscript and/or data are unclear and a few experimental points that will require attention.

1. The data presented are consistent with the (very interesting) model at the end. However, there are some experiments that should be done to firm up this conclusion. In particular, the model postulates that the normal role of the Toll pathway in this context is, ultimately, to enable brummer-mediated triglyceride catabolism in the intestine. If this is the case, uncomplicated NP1-driven brummer knockdown should phenocopy the data shown for seml and dif with regard to intestinal bacteria. If this were shown, it would lend credence to the model as a whole.

2. The triglyceride measurements are unclear. Do they show whole-fly triglyceride (as implied by the graph in the figure) or intestinal triglyceride (as implied by the text)? In this context, it’s important to show intestinal triglyceride specifically. Given the sometimes ambiguous interpretation of commercial triglyceride assays like that used here, the authors should verify this reading by staining triglyceride in dissected intestines with a neutral lipid stain (Oil Red O, for example, is inexpensive and works well).

3. It looks as though all experiments were done in females, except the triglyceride measurement, which was done in males. Please show the female data throughout—there are big differences between the sexes in intestinal physiology and switching back and forth between them can be misleading.

4. Some of the experiments are missing controls. In particular, there are no positive controls (wild-type) shown in Figure 3A or Figure 5. Wild-type samples done in parallel should be shown.

5. The quantification in Figure S3 appears to be a single experiment. Where did the significance measurements come from?

6. The introduction is really absurdly long, but it doesn’t cover the really relevant literature on Vibrio cholerae use of Drosophila intestinal triglyceride (Kamareddine et al. Nature Microbiology 3: 243–252 and Liimatta et al. Applied and Environmental Microbiology 84: e01113-18).

Minor points

7. The graph labels in Figure 5C must be wrong?

8. The methods don’t include details of the genetic introgression of the dif and seml mutants onto the DGRP line.

Reviewer #2: The manuscript entitled “Bacterial recognition and Toll signalling sustain commensal gut bacteria in Drosophila” describes the identification of the role of PGRP-SA receptor protein in the maintenance of commensal bacteria in drosophila gut through control of lipid metabolism. Interaction between commensal bacteria and the host physiology and especially immune system is the focus of intense research in the drosophila and mammals model systems. Several papers addressed the action of gut microbiota on gut morphology and the modification of gut metabolism in response to infection. In the present manuscript, the authors address the question of the maintenance of gut microbiota through interaction with the gut and propose that bacteria are sensed by PGRP-SA which allows, via control of TOR and 4EF-BP, allocation of lipids to the gut lumen favorizing the maintenance of commensal gut bacteria. However, this model is far from being demonstrated by the presented work.

The main concern is related to the article of Vandehoef et al published in Cell Reports 2020. This article clearly shows that the NF-kB/Relish host signaling function in midgut enterocytes is vital to adapt gut microbiota species abundance and composition to host diet

macronutrient composition through the control of 4E-BP/Thor and cap-dependent translation. This reference is only cited in the introduction (line 99) without any mention of Relish. Given the extensive similarity on part of the data between this paper and the present submitted manuscript (control of gut microbiota by a NFkB factor through the activity of 4E-BP), this should absolutely be discussed in the present manuscript. Furthermore, the authors make no reference to Molaei et al published in Developmental Cell 2019 that links Relish and Blummer for the control of lipolysis in the fat body. One important point raised by the articles is the involvement of Relish, the transcription factor of the IMD pathway in the control of gut microbiota. Relish and the IMD pathway are known to be active in the gut, activated by commensal bacteria, which has not been shown for PGRP-SA and the Toll pathway. As a consequence, everything that is described in the present manuscript could be due to Relish activation and this should absolutely be tested. Given that the involvement of the Toll pathway is not demonstrated (see below) and that there are questions about the expression of PGRP-SA in the gut, this manuscript can’t be accepted without clarification of this major point: the possibility that PGRP-SA activates Relish in the gut, the verification that PGRP-SA is really expressed in the gut. If Relish is indeed activated downstream of PGRP-SA, then the whole part concerning 4E-BP and Blummer is already published in Vandehoef et al. The focus of this work would be to understand how PGRP-SA sense bacteria and activate Relish.

Before going to specific points, I would like to say that the logic of the text is quite difficult to follow. It would need a substantial work in order to clarify either the rational for some experiments or the conclusions raised from data. Furthermore, the manuscript contains a substantial number of mistakes that could have easily corrected before submission.

The title is clearly not supported by the data:

There is no demonstration that bacteria are sensed, except by the fact that PGRP-SA is a circulating receptor shown to activate the Toll pathway. But PGRP-SA was described to be secreted in the hemolymph and we don’t know is PGRP-SA could be found in the gut lumen. In the various data accessible on Flybase, PGRP-SA has not been shown to be expressed in the gut. There is one experiment showing that its inactivation specifically in the gut gives the same phenotype as inactivation in the whole flies. To clearly say that bacteria are sensed by PGRP-SA, it requires to demonstrate that PGRP-SA is indeed expressed in the gut. If bacteria sensing is involved, axenic flies should have an accumulation of intestinal fat. Is it the case?

The demonstration that PGRP-SA could indeed bind to some of the commensal bacteria would be nice but probably a bit difficult.

Toll signaling is not demonstrated: the loss of PGRP-SA could result in the loss of Toll signaling, but it is not proved. There are no data showing that Toll signaling is activated in the gut. The authors mentioned drs-GFP flies in the method but they have not been used. Is there an expression in the gut? Depending on bacterial load? Is it diminished in axenic flies? The only other component of the pathway tested is DIF and only in figure 1 for its effect on bacterial load and its inactivation was not specifically in the gut. The authors also show that the life span of dif mutants is shorten but surprisingly it is the only experiment in which they don’t test PGRP-SA mutants.

The authors proposed that fat catabolism down-regulation is the explanation for the lack of commensal bacteria maintenance. The rational for testing intestinal fat level is not explained. Most importantly, I’m not sure whether they really analyzed fat level in the gut or in whole flies since the text and figure legend state that they analyzed intestinal fat level but the figure and the material and methods described whole fly fat level analysis. They should clarify this and test is lipid droplets accumulate in the gut epithelium.

The authors proposed that Brummer (bmm) could be implicated in fat catabolism. They should explain in the text the function of Bmm. It’s function in the gut was not described yet. Therefore, they should test if its knock-down in the gut impacts intestinal fat level, but also commensal bacteria density. They should also test intestinal fat level in yw seml ; uas bmm RNAI flies treated with rapamycin.

The fact that in 30-day old seml flies, the bacterial density is similar to wild-type levels is quite intriguing. The authors suggest that this is due to bacterial defecation and re-eating. But what makes the gut different that bacteria can live in the gut at 30 days when they were unable at 5 days? Could they verify that these old flies have indeed an increased intestinal fat level to rule out that they adapted their intestinal metabolism?

Another point is that in Fig2C they show that in 30-day old yw,seml flies commensal diversity is more similar to 5-day than 30-day old yw flies commensal diversity. Can we suppose that re-eating keeps the commensal in a young state (early colonization of the gut)? The authors should at last comment on that. Putting axenic flies on medium where old flies defected for 24h for example should lead to the same commensal diversity as 30-day old seml flies.

Gut microbiota is known to stimulate proliferation of intestinal stem cells. However, figure S1 does not show that there is less proliferation, as stated by the authors (line 209-210) but that there is less ISC and EB progenitors, even if it was published that a small decrease of esg-GFP+ cells is correlated with a reduction of proliferative cells. What is more problematic is that in material and methods, line 662, the authors say that the anterior gut was imaged whereas Broderick et al 2014 showed that it is in the posterior part of the gut that the lack of commensal induces a reduction in esg>GFP staining and proliferation.

Is there a consequence of this lack of proliferation on the gut morphology?

There is a confusion in the text between intestinal stem cells and progenitors (enteroblasts). They also say (line 208) that, in mammals, Paneth cell are the progenitors or the stem cells of enterocytes. That is not true: stem cells are LGR5+ cells in between Paneth cells.

I don’t understand the rational for the experiment presented in Fig5B. The authors explain that NF-kB signaling could regulate 4E-BP expression but dTOR regulates 4E-BP activity through its phosphorylation but the authors are looking at the expression of the gene in PGRP-SA and dTOR knock-down flies. It is possible to test 4E-BP phosphorylation. However, it was shown that Relish is able to regulate 4E-BP expression. If DIF is really involved in this process, it should be tested that DIF is also able to activate the expression of 4-BP.

Other more specific questions:

Line 68: please be more precise about the meaning of « host signaling »

Line 320-333: there should be a comment on the fact that rapamycin treated seml flies have a similar bacterial diversity than rapamycin treated flies and therefore that rapamycin treatment is dominant over the seml/wt genetic difference.

Fig 2B: was the replicate included in this analysis? Figure 2 and S4 could me mixed.

Fig S3: how many experiments were done? There is no indication of variation between experiments.

Fig3A: wt control is missing to see whether the bacterial density is back to normal.

Line 361: be more precise than “regulating gut bacteria”. I guess you mean bacterial composition of microbiota.

Line 362: the sentence is unclear: it is 4E-BP coding gene that includes NF-kB binding sites.

Line 364: it is unclear what is corroborated by the induction of 4E-BP by C. albicans.

Figure S5 is missing, replaced by another version of S6.

Fig S7: the name Thor is used for 4E-BP. Since this name is also used in the references cited in the text. The corresponding text uses 4E-BP which is confusing. It would help to use the same name in the whole manuscript.

Fig4A: the authors show the control data to help understand the figure and show variability.

Fig5C: annotation of the figure is wrong (TOR instead of 4E-BP).

FigS8 and FigS9 have been exchanged.

FigS9 and line 392: there is no comment on the difference between males and females. In females the reduction of bacterial load is quite low. Is the difference between wild-type males and females significative?

**Have all data underlying the figures and results presented in the manuscript been provided?**

Reviewer #1: Yes

Reviewer #2: None

PLOS authors have the option to publish the peer review history of their article (what does this mean?). If published, this will include your full peer review and any attached files.

Reviewer #1: No

Reviewer #2: No

---

## [Decision Letter · Decision Letter 1]

14 Dec 2021

Dear Dr Ligoxygakis,

We are pleased to inform you that your manuscript entitled "Bacterial recognition by PGRP-SA and downstream signalling by Toll/DIF sustain commensal gut bacteria in Drosophila." has been editorially accepted for publication in PLOS Genetics. Congratulations!

Yours sincerely,

Gregory P. Copenhaver

Editor-in-Chief

PLOS Genetics

Comments from the reviewers (if applicable):

Reviewer's Responses to Questions

**Comments to the Authors:**

Reviewer #1: I'm satisfied that most of my requests have been met. There's a good deal of new data in this version, which adds a significant amount of value.

**Have all data underlying the figures and results presented in the manuscript been provided?**

Reviewer #1: None

PLOS authors have the option to publish the peer review history of their article (what does this mean?). If published, this will include your full peer review and any attached files.

Reviewer #1: No

**Data Deposition**

http://datadryad.org/submit?journalID=pgenetics&manu=PGENETICS-D-21-00436R1

**Press Queries**

---

## [Editor Report · Acceptance letter]

5 Jan 2022

PGENETICS-D-21-00436R1 

Bacterial recognition by PGRP-SA and downstream signalling by Toll/DIF sustain commensal gut bacteria in Drosophila. 

Dear Dr Ligoxygakis, 

We are pleased to inform you that your manuscript entitled "Bacterial recognition by PGRP-SA and downstream signalling by Toll/DIF sustain commensal gut bacteria in Drosophila." has been formally accepted for publication in PLOS Genetics! Your manuscript is now with our production department and you will be notified of the publication date in due course.

With kind regards,

Anita Estes

PLOS Genetics

On behalf of:
